# CONFIDENCE ESTIMATION USING UNLABELED DATA

**Chen Li** *
Stony Brook University

**Xiaoling Hu**
Stony Brook University

**Chao Chen**
Stony Brook University

## ABSTRACT

Overconfidence is a common issue for deep neural networks, limiting their deployment in real-world applications. To better estimate confidence, existing methods mostly focus on fully-supervised scenarios and rely on training labels. In this paper, we propose the first confidence estimation method for a semi-supervised setting, when most training labels are unavailable. We stipulate that even with limited training labels, we can still reasonably approximate the confidence of model on unlabeled samples by inspecting the prediction consistency through the training process. We use training consistency as a surrogate function and propose a consistency ranking loss for confidence estimation. On both image classification and segmentation tasks, our method achieves state-of-the-art performances in confidence estimation. Furthermore, we show the benefit of the proposed method through a downstream active learning task.

## 1 INTRODUCTION

Besides accuracy, the confidence, measuring how certain a model is of its prediction, is also critical in real world applications such as autonomous driving (Ding et al., 2021) and computer-aided diagnosis (Laves et al., 2019). Despite the strong prediction power of deep networks, their overconfidence is a very common issue (Guo et al., 2017; Nguyen et al., 2015; Szegedy et al., 2014). The output of a standard model, e.g., the softmax output, does not correctly reflect the confidence. The reason is that the training is only optimized regarding to the training set (Naeini et al., 2015), not the underlying distribution. Accurate confidence estimation is important in practice. In autonomous driving and computer-aided diagnosis, analyzing low confidence samples can help identify subpopulations of events or patients that deserve extra consideration. Meanwhile, reweighting hard samples, i.e., samples on which the model has low confidence, can help improve the model performance. Highly uncertain samples can also be used to promote model performance in active learning (Siddiqui et al., 2020; Moon et al., 2020).

Different ideas have been proposed for confidence estimation. Bayesian approaches (MacKay, 1992; Neal, 1996; Graves, 2011) rely on probabilistic interpretations of a model's output, while the high computation demand restricts their applications. Monte Carlo dropout (Gal & Ghahramani, 2016) is introduced to mitigate the computation inefficiency. But dropout requires sampling multiple model predictions at the inference stage, which is time-consuming. Another idea is to use an ensemble of neural networks (Lakshminarayanan et al., 2017), which can still be expensive in both inference time and storage. To overcome the inefficiency issue, some recent works focus on the whole training process rather than the final model. Moon et al. (2020) use the frequency of correct predictions through the training process to approximate the confidence of a model on each training sample. Geifman et al. (2018) collect model snapshots over the training process to compensate for overfitting and estimate confidence.

However, most existing methods purely rely on labeled data, and thus are not well suited for a semi-supervised setting. Indeed, confidence estimation is critically needed in the semi-supervised setting, where we have limited labels and a large amount of unlabeled data. A model trained with limited labels is sub-optimal. Confidence will help efficiently improve the quality of the model, and help annotate the vast majority of unlabeled data in a scalable manner (Wang et al., 2022; Sohn et al., 2020; Xu et al., 2021).

In this paper, we propose the first confidence estimation method specifically designed for the semi-supervised setting. The first challenge is to leverage the vast majority of unlabeled data for confi-

---

*Email: Chen Li (li.chen.8@stonybrook.edu).

dence learning. For data without labels, our idea is to use the consistency of the predictions through the training process. An initial investigation suggests that consistency of predictions tends to be correlated with sample confidence on both labeled and unlabeled data.

Having established training consistency as an approximation of confidence, the next challenge is that the consistency can only be evaluated on data available during training. To this end, we propose to re-calibrate model's prediction by aligning it with the consistency. In particular, we propose a novel *Consistency Ranking Loss* to regulate the model's output after the softmax layer so it has a similar ranking of model confidence output as the ranking of the consistency. After the re-calibration, we expect the model's output correctly accounts for its confidence on test samples. We both theoretically and empirically validate the effectiveness of the proposed Consistency Ranking Loss. Specifically, we show the superiority of our method on real applications, such as image classification and medical image segmentation, under semi-supervised settings. We also demonstrate the benefit of our method through active learning tasks.

**Related work.** There are two mainstream approaches for confidence (uncertainty) estimation: confidence calibration and ordinal ranking. Confidence calibration treats confidence as the true probability of making correct prediction and tries to directly estimate it (Platt, 2000; Guo et al., 2017; Jungo & Reyes, 2019; Zadrozny & Elkan, 2002; Naeini et al., 2015). For any sample, the confidence estimate generated by a well-calibrated classifier should be the likelihood of predicting correctly. Directly estimating the confidence may be challenging. Instead, many works focus on the ordinal ranking aspect (Geifman et al., 2018; Geifman & El-Yaniv, 2017; Mandelbaum & Weinshall, 2017; Moon et al., 2020; Lakshminarayanan et al., 2017). In spite of the actual estimated confidence values, the ranking of samples with regard to the confidence level should be consistent with the chance of correct prediction. A model with well-ranked confidence estimate can be widely used in the field of selective classification, active learning and semi-supervised learning (Siddiqui et al., 2020; Yoo & Kweon, 2019; Sener & Savarese, 2018; Tarvainen & Valpola, 2017; Zhai et al., 2019; Miyato et al., 2018; Xie et al., 2020; Chen et al., 2021; Li & Yin, 2020). Semi-supervised active learning methods (Gao et al., 2020; Huang et al., 2021) only focus on finding high-uncertainty samples through training actions, thus are unable to be applied to estimate uncertainty for unseen samples.

## 2 CONSISTENCY - A NEW SURROGATE OF CONFIDENCE

Our main idea is to use the training consistency, i.e., the frequency of a training datum getting the same prediction in sequential training epochs, as a surrogate function of the model's confidence. In this section, we first formalize the definition of training consistency. Next, we use qualitative and quantitative evidences to show that consistency can be used as a surrogate function of confidence. This justifies the usage of training consistency as a supervision for model confidence estimation, which we will introduce in the next section.

**Definition: training consistency.** Assume a given dataset with $n$ labeled and $p$ unlabeled data, $\mathcal{D} = (X, Y, U)$. Here $X = \{x_1, \ldots, x_n\}, x_i \in \mathcal{X}$ is the set of labeled data with corresponding labels $Y = \{y_1, \ldots, y_n\}, y_i \in \mathcal{Y} = \{1, \ldots, K\}$. The set of unlabeled data $U = \{x_{n+1}, \ldots, x_{n+p}\}, x_j \in \mathcal{X}$ cannot be directly used to train the model, but will be used to help estimate confidence. We assume a simple training setting where we use the labeled set $(X, Y)$ to train a model $f(x, y; W) : \mathcal{X} \times \mathcal{Y} \to [0, 1]$. The method can naturally generalize to more sophisticated semi-supervised learning methods, where unlabeled data can also be used. For any data, either labeled or unlabeled, $x_i \in X \cup U$, we have the classification $\hat{y}_i = \arg\max_{y \in \mathcal{Y}} f(x_i, y; W)$. Note that traditionally the model output regarding the classification label, $f(x_i, \hat{y}_i; W) = \max_{y \in \mathcal{Y}} f(x_i, y; W)$, is used as the confidence.

Our definition involves the training process. Denote by $W^t$ the model weights at the $t$-th training epoch, $t = 1, \ldots, T$. We use $\hat{y}_i^t = \arg\max_{y \in \mathcal{Y}} f(x_i, y | W^t)$ to denote the model classification for sample $x_i$ at the $t$-th epoch. For a sample $x_i$, we define its *training consistency* as the frequency of getting consistent predictions in sequential training epochs during the whole training process ($T$ epochs in total):

$$c_i = \frac{1}{T-1} \sum_{t=1}^{T-1} \mathbb{1}\{\hat{y}_i^t = \hat{y}_i^{t+1}\}. \tag{1}$$

**Qualitative analysis shows consistency is a good surrogate of confidence.** We provide a qualitative example in Fig. 1 with the feature representations. We observe that the data further from the

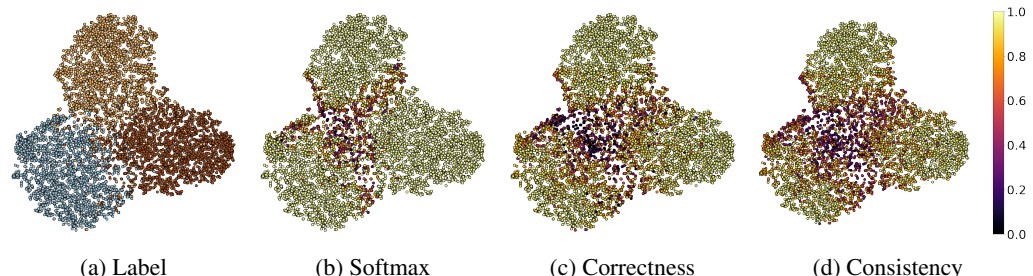

(a) Label          (b) Softmax          (c) Correctness          (d) Consistency

Figure 1: Feature representations of 6000 unlabeled CIFAR10 data. The model is trained on 20% labeled data. We use t-SNE for dimension reduction and only show data of three classes. (**a**) different class labels: 0-blue, 1-yellow, 9-brown. (**b**) the softmax output is clearly overconfident. (**c**) and (**d**) are correctness and consistency maps respectively. They are similar to each other and seem to be well correlated with confidence.

decision boundary have higher consistency, and the ones closer to decision boundary have lower consistency. This is consistent with our intuition and seems to correlate with model confidence. Although we do not have the ground truth of confidence, it is reasonable to believe that data further from the decision boundary will have higher confidence, whereas data near the decision boundary will have lower confidence.

The data representation is acquired by training a model (PreAct-ResNet110) on only $20\%$ of CI-FAR10 training set. We extract the penultimate fully connected (FC) layer output as the feature representation. Here we use t-SNE to visualize the feature representations of $80\%$ unlabeled training images. For ease of exposition, we only focus on class 0, class 1 and class 9. As shown in Fig. 1(a), since the model is trained with limited labeled data, the three classes are not completely separated in the representation space. We expect lower confidence data to be closer to the decision boundary. However, this is not well reflected by the model confidence output, i.e., the maximum softmax output, $f(x_i, \hat{y}_i; W)$. As shown in Fig. 1(b), softmax output tends to be overconfident.

Meanwhile, we calculate consistency on unlabeled data and visualize them in Fig. 1(d). The consistency map seems much more reasonable and better correlates with model confidence. The consistency is high within each class, but slowly transits to low towards the decision boundary. Since we do not have ground truth of confidence, as a reference, we also calculate and visualize the *correctness* of data, a known good proxy of confidence in fully-supervised setting (Moon et al., 2020). The correctness of a datum $x_i$ is the frequency at which the model makes a correct classification on $x_i$ across different epochs, i.e., $\frac{1}{T}\sum_{t=1}^{T} \mathbb{1}\{y_i = \hat{y}_i^t\}$. As shown in Fig. 1(c), the correctness map is very similar to consistency. This further justifies our proposal of using consistency.

Please note that this analysis does not justify correctness as a good confidence surrogate when training with limited labels. When labeled data is limited, we can only calculate and use correctness on a limited amount of samples. They cannot provide sufficient supervision for confidence learning. This will be evident in our experiment section.

**Quantitative analysis.** We further evaluate the quality of consistency as a confidence surrogate using a popular confidence evaluation metric, the area under the risk-coverage curve (AURC) and the excess-AURC (E-AURC). Based on the ordinary ranking principle, AURC and E-AURC measure how well an estimated confidence is correlated with the chance of correct prediction; a sample with high confidence should have a high chance to be correctly predicted. A low AURC or E-AURC indicates the estimation is a good approximation of the true confidence. A formal definition of AURC and E-AURC is provided in Sec. 3.2.

We evaluate AURC and E-AURC on the training data for consistency, correctness and softmax output of the model at different training epochs with regard to a model trained on limited labeled data. The setting is the same as in the previous qualitative analysis. The results are shown in Fig. 2. We observe that overall all three confidence proxies improve their quality as the training continues (i.e., their AURCs/E-AURCs decrease as the training progresses). Consistency is better than softmax output throughout the training (i.e., having a lower AURC or E-AURC). Correctness has the best performance among the three (i.e., the lowest AURC or E-AURC). However, correctness is unavailable for unlabeled data, thus does not help learning confidence in our problem. In Append. A.8, we compare training consistency with other label-free uncertainty surrogates.

## 3 LEARNING TO ESTIMATE CONFIDENCE: CONSISTENCY RANKING LOSS

We have established that consistency is a good approximation of the confidence. However, consistency is only available for training data. At the inference stage, we do not have such measure for test data. To this end, we propose a novel Consistency Ranking Loss during training to enforce the model's maximum softmax output approximates the consistency well. The hope is at the inference stage, the maximum softmax output will well approximate the consistency, and thus the confidence.

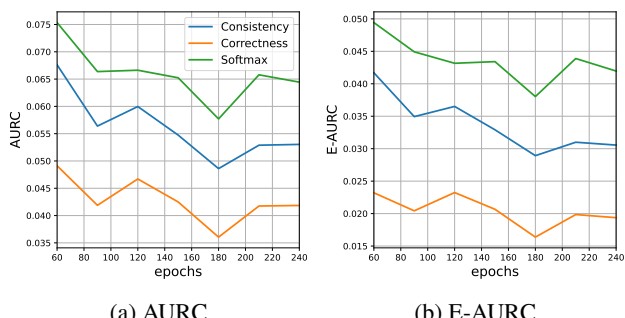

(a) AURC          (b) E-AURC

Figure 2: Performance of consistency as a confidence estimator. We evaluate the performance of softmax output, consistency and correctness in terms of popular metrics, AURC and E-AURC. We train a model with only 20% of CIFAR10 labeled data, and evaluate on the remaining unlabeled data.

The consistency ranking loss is a pairwise ordinal ranking function to enforce the maximum softmax output to be consistent with the training consistency on all labeled and unlabeled training data $X \cup U$. Recall for the $i$-th training datum, $x_i \in X \cup U$, we denote by $c_i$ its training consistency (Eq. (1)). For convenience, we denote by $\kappa_i$ the model's maximum softmax output for $x_i$.[1] The loss function enforces the two to be consistent in terms of ordinary ranking, i.e., if $c_i < c_s$, then $\kappa_i < \kappa_s$. Formally,

$$\mathcal{L}_{cons}(f) = \sum_{s=1}^{n+p} \sum_{i=1, c_i < c_s}^{n+p} \max\{0, (c_s - c_i) - (\kappa_s - \kappa_i)\}. \tag{2}$$

The loss sums over all pairs of training data and penalizes a pair $(i, s)$ when $x_i$ has a lower confidence than $x_s$ and the difference $c_s - c_i$ is bigger than the difference $\kappa_s - \kappa_i$. Through training, the confidence estimator $\kappa$ is learned to have a similar gap as the consistency $c$'s, i.e., $\kappa_s - \kappa_i \geq c_s - c_i$, $\forall c_s > c_i$. The loss seems expensive as it is quadratic to the sample size. In practice, however, computation is not a major issue. We evaluate the loss on samples within each minibatch (both labeled and unlabeled). The loss is then only quadratic to the minibatch size. As shown in the experiment section, this suffices in practice. The details of the pairing strategy are provided in Append. A.2.

To fully exploit labeled training samples for confidence estimation, we also incorporate correctness into our training, using a ranking loss $\mathcal{L}_{corr}$. Note $\mathcal{L}_{corr}$ is only evaluated on labeled samples. Our overall loss $\mathcal{L}$ is

$$\mathcal{L} = \mathcal{L}_{CE} + \lambda_1 \mathcal{L}_{corr} + \lambda_2 \mathcal{L}_{cons} \tag{3}$$

Here $\mathcal{L}_{CE}$ is cross entropy loss on labeled samples. $\lambda_1$ and $\lambda_2$ are the weights of losses. The pseudo code of our method is shown in Append. A.6

### 3.1 EXPERIMENT VERIFICATION

In practice, it is intractable to compare all possible sample pairs even for small batch data. We use experiment to show the empirical performance of consistency ranking loss on CIFAR-10 dataset. Here we adopt the same experimental setting as Sec. 2. All the results are collected from unlabeled training set (40000). As shown in Tab. 1, our consistency ranking loss makes the AURC (E-AURC) of model confidence output (maximum softmax output) approximate the AURC (E-AURC) of training consistency very well. The experimental results indicate that our consistency ranking loss can achieve very good performance even optimized on minibatch data with limited pairs.

### 3.2 THEORETICAL GUARANTEE

In this section, we provide theoretical analysis of the proposed consistency ranking loss (Eq. (2)). We show that the loss upperbounds the quality of the confidence estimator $\kappa$. Informally, our main

---

[1] In general, $\kappa_i$ can be any confidence estimator (e.g., one can use a separate head to output $\kappa_i$).

Table 1: The quality of optimization on consistency ranking loss. Softmax (ours): the maximum softmax output trained with consistency ranking loss. TC (ours): training consistency. Diff. (ours): the evaluation metric difference between our maximum softmax output and training consistency. Softmax, TC, Diff. are corresponding results without consistency ranking loss. All values $\times 10^2$.

| Dataset (labeled size) | Method | E-AURC↓ | AURC↓ | Method | E-AURC↓ | AURC↓ |
|---|---|---|---|---|---|---|
| CIFAR10 (10000) | Softmax | 29.07 | 40.57 | Softmax (ours) | 19.35 | 32.57 |
| | TC | 17.63 | 29.13 | TC (ours) | 17.17 | 30.39 |
| | Diff. | 11.44 | 11.44 | Diff. (ours) | **2.17** | **2.17** |

theorem (Thm. 1) states that the loss upperbounds the difference between the confidence estimation performance of $\kappa$ and the consistency $c$, in terms of the popular metrics AURC and E-AURC. In other words, as we minimize the loss, $\kappa$ will have a similar performance as the consistency $c$ in terms of confidence estimation.

**Definition of AURC.** To evaluate if a confidence estimator $\kappa$ is correlated with the chance of a sample being correctly predicted by classifier $f$, the area under the risk-coverage curve (AURC) is proposed. Given a dataset with $m$ samples $\mathcal{D}_m = (X_m, Y_m)$ and a confidence estimator $\kappa$, the empirical selective risk is $\hat{r}(f, g_\theta | \mathcal{D}_m) = \frac{\frac{1}{m} \sum_{i=1}^{m} l(\hat{y}_i, y_i) g_\theta(x_i)}{\frac{1}{m} \sum_{i=1}^{m} g_\theta(x_i)}$ where $\hat{y}_i = \arg\max_{y \in \mathcal{Y}} f(x_i, y | \mathcal{D}_m)$. We define indicator function $l(\hat{y}_i, y_i) = \mathbb{1}\{\hat{y}_i \neq y_i\}$. Here, $g_\theta(x_i) = \mathbb{1}\{\kappa_i \geq \theta\}$ is a selection function thresholding on confidence scores $\kappa$. The AURC is defined as $AURC(f, \kappa | \mathcal{D}_m) = \frac{1}{m} \sum_{\theta \in \Theta} \hat{r}(f, g_\theta | \mathcal{D}_m)$, where $\Theta$ is the set of all distinct values of confidence scores $\kappa$ on $X_m$, i.e., $\Theta = \{\kappa_i | i = 1, \ldots, m\}$.

E-AURC was proposed as a normalized version of AURC. $\kappa_*$ is an optimal confidence score function of $f$ on $\mathcal{D}_m$, yielding the correctly predicted samples having higher confidence scores than misclassified ones. $\kappa_*$ minimizes $AURC(f, \cdot | \mathcal{D}_m)$. Excess-AURC (E-AURC) is defined as, $E\text{-}AURC(f, \kappa | \mathcal{D}_m) = AURC(f, \kappa | \mathcal{D}_m) - AURC(f, \kappa_* | \mathcal{D}_m)$. When $E\text{-}AURC(f, \kappa | \mathcal{D}_m) = 0$ and $f$ is not an optimal classifier, then $\kappa$ assigns higher confidence score to correct predictions than incorrect ones. In such a way, we can filter out correct predictions by $\kappa$.

Next, we formally state and prove our main theorem. Let $T(f | \mathcal{D}_m)$ be the set constituted by all the correctly predicted samples, $T(f | \mathcal{D}_m) = \{x : \hat{y} = y, \hat{y} = \arg\max_{i \in \mathcal{Y}} f(x)_i, (x, y) \in \mathcal{D}_m\}$. Here we define $R_\kappa$, a ranking function based on confidence score function $\kappa$ on $X_m$, $R_\kappa(x) \in \{1, 2, \ldots, m\}, \forall x \in X_m$. If $\kappa(x_i) < \kappa(x_j)$ than $R_\kappa(x_i) > R_\kappa(x_j)$, $\forall x_i, x_j \in X_m$. Here we assume both $\kappa$ and $R_\kappa$ are one-to-one functions. Given $R_\kappa$, we define ranking set $H_\kappa = \{R_\kappa(x) : \hat{y} = y, \hat{y} = \arg\max_{i \in \mathcal{Y}} f(x)_i, (x, y) \in \mathcal{D}_m\}$. Here we define $c(x_i) = c_i, x_i \in X_m$. Note that training consistency is a kind of confidence estimator and in the context below, we will use $c$ to denote training consistency. Here we define $R_c$ in certain way such that it is one-to-one function (More details are included in Append. A.3).

**Definition 3.1.** Given ranking function $R_c$ and $R_\kappa$, we introduce a set $D_c = \{|c(R_c^{-1}(i)) - c(R_c^{-1}(j))|; i = \gamma_m(R_\kappa(x)), j = \gamma_m(R_\kappa(x)) + sign(R_\kappa(x) - \gamma_m(R_\kappa(x)))), x \in T, R_c^{-1}(j) \notin T(f | \mathcal{D}_m)\}$, where $\gamma_m = \arg\min_{\gamma \in \tau(H_\kappa, H_c)} \sum_{x \in T} |R_\kappa(x) - \gamma(R_\kappa(x))|$, $sign$ is a sign function, $\tau$ is the collection of the bijective function between the elements of $H_\kappa$ and $H_c$.

**Theorem 1.** *Given dataset $\mathcal{D}_m = (X_m, Y_m)$ and training consistency $C_m$:*

$$|AURC(f, c | \mathcal{D}_m) - AURC(f, \kappa | \mathcal{D}_m)| \leq \frac{1}{m \min D_c} \mathcal{L}_{cons}(f, X_m; C_m) \qquad (4)$$

We notice that $|E\text{-}AURC(f, c | \mathcal{D}_m) - E\text{-}AURC(f, \kappa | \mathcal{D}_m)| = |AURC(f, c | \mathcal{D}_m) - AURC(f, \kappa | \mathcal{D}_m)|$, thus the theorem above works for *E-AURC* too.

*Proof.* Let $T_\kappa(i)$ be the set constituted by all the correct predicted samples with confidence less or equal than $R_\kappa^{-1}(i)$, $T_\kappa(i) = \{x : R_\kappa(x) \leq i, \hat{y} = y, \hat{y} = \arg\max_{i \in \mathcal{Y}} f(x)_i, (x, y) \in \mathcal{D}_m\}$. $T_c(i)$ is the corresponding set introduced by training consistency $c$. We define $H_c(f, \kappa | \mathcal{D}_m) = \sum_{i=1}^{m} \mathbb{1}(\#T_\kappa(i) \neq \#T_c(i))$ and $H_c^k(f, \kappa | \mathcal{D}_m) = \sum_{i=1}^{k} \mathbb{1}(\#T_\kappa(i) \neq \#T_c(i))$. It can be proved:

$$|AURC(f, c | \mathcal{D}_m) - AURC(f, \kappa | \mathcal{D}_m)| \leq \frac{H_c(f, \kappa | \mathcal{D}_m)}{m} \qquad (5)$$

There is a fact that $\forall i \in \{1, 2, \ldots, m\}$ if $\mathbb{1}(\#T_\kappa(i) = \#T_c(i))$, than $\hat{r}(f, g_{R_\kappa^{-1}(i)}|\mathcal{D}_m) = \hat{r}(f, g_{R_c^{-1}(i)}|\mathcal{D}_m)$, so $|AURC(f, c|\mathcal{D}_m) - AURC(f, \kappa|\mathcal{D}_m)| \leq \frac{1}{m}\sum_{\#T_\kappa(i)\neq\#T_c(i)} |\hat{r}(f, g_{R_\kappa^{-1}(i)}|\mathcal{D}_m) - \hat{r}(f, g_{R_c^{-1}(i)}|\mathcal{D}_m)|$. We also have $F_r(f, c|\mathcal{D}_m) = \sum_{i=1}^m \prod_{j=1}^m \mathbb{1}(R_c(x_j) \leq i) \cdot \mathbb{1}(\hat{y}_j = y_j)$, where $\hat{y}_j = \arg\max_{i\in\mathcal{Y}} f(x_j)_i$ and $(x_j, y_j) \in \mathcal{D}_m$. Here we define $K(f, c|\mathcal{D}_m) = \#T(f|\mathcal{D}_m) - F_r(f, c|\mathcal{D}_m)$, which is the number of correct predicted samples with lower training consistency than at least one incorrect sample. Then we prove that:

$$\frac{1}{m \min D_c}\mathcal{L}_{cons}(f, X_m; C_m) \geq \frac{H_c(f, \kappa|\mathcal{D}_m)}{m} \tag{6}$$

A detailed proof version is provided in Append. A.3. □

## 4 EXPERIMENT

To verify the effectiveness of our method in confidence estimation with limited labeled data, we start with evaluating on image classification and medical image segmentation tasks. Active learning is usually used as a follow-up task of confidence estimation. Here we implement active learning on image classification tasks to show that the highly uncertain samples picked up by our method will promote the learning process in a better manner. We also evaluate our method on CIFAR10/100 through anomaly detection in Append. A.7.

### 4.1 IMAGE CLASSIFICATION

In this section, we show that our method can generate reliable confidence estimates with a small portion of labeled samples and majority of unlableled samples through classification tasks. We evaluate our method on benchmark datasets, CIFAR-10 and CIFAR-100 (Krizhevsky et al., 2009). Besides, we also provide the results on cancer survival dataset in Append. A.9.

**Implementation details.** To show the superiority of our method in making use of unlabeled training samples, we compare our method with other baselines under different portions of labeled training data: 5% (2500), 10% (5000), 20% (10000), 100% (50000). As for network architecture, we adopt PreAct-ResNet110 (He et al., 2016) for CIFAR-10 and DenseNetBC ($k = 12$, $d = 100$) (Huang et al., 2017) for CIFAR-100. All methods are trained by SGD with a momentum of 0.9 and a weight decay of 0.0001. We train our method for 300 epochs with the mini-batch size of 192, in which 64 are labeled, and use initial learning rate of 0.1 with a reduction by a factor of 10 at 150 and 250 epochs. A standard data augmentation scheme for image classification is used, including random horizontal flip, 4 pixels padding and $32 \times 32$ random crop. More details are included in Append. A.1.

For our method, we set the loss weights $\lambda_1, \lambda_2$ in Eq. 3 as $0.5, 0.5$ respectively, and use maximum softmax as the model confidence output for both training and evaluation. We compare our method with cross entropy loss only (Softmax), CRL (Moon et al., 2020), AES with 30 snapshot models (Geifman et al., 2018), Mcdrop with 50 stochastic ensembles (Gal & Ghahramani, 2016) and Aleatoric+MCdropout (Kendall & Gal, 2017). As for evaluation, besides the ordinal ranking evaluation metrics AURC and E-AURC, we also adopt three widely used calibration evaluation metrics, the expected calibration error (ECE) (Naeini et al., 2015), negative log likelihood (NLL) and the Brier score (Brier et al., 1950). The false positive rate at 95% true positive rate (FPR-95%-TPR) is also used to evaluate confidence estimation performances.

The experimental results are shown in Tab. 2 (The results of 'full' setting are included in Append. A.5). All the experiments are repeated for five times, and we report the means and standard deviations. From the results, we can see that the proposed method achieves better performances in both confidence estimation and classification accuracy. The improvement of our consistency ranking loss is even more salient under the scenarios of less labeled training samples. This indicates that our method utilizes unlabeled samples to encourage neural network classifier to generate believable confidence estimates effectively.

It is also obvious that the improvement of our method is more significant on CIFAR-100. This is because for CIFAR-100, the number of training samples in each class is much less than CIFAR-10, which worsens 'the lack of samples' effect in confidence estimation and classification. This further demonstrates the necessity of utilizing the confidence information of unlabeled samples .

Table 2: Comparison of confidence estimates on CIFAR-10/100 with various labeled training data sizes. The best results are shown in bold. We multiply the values of AURC & E-AURC by $10^3$, FPR by $10^2$ and NLL by 10 for clarity.

| Dataset (labeled size) | Method | Acc↑ | AURC↓ | E-AURC↓ | FPR-95↓ | ECE↓ | NLL↓ | Brier↓ |
|---|---|---|---|---|---|---|---|---|
| | | | | CIFAR10 | | | | |
| | Softmax | 0.717±0.008 | 109.78±6.69 | 65.32±3.92 | 75.64±0.59 | 20.23±0.76 | 14.83±0.60 | 47.17±1.53 |
| | AES | 0.713±0.002 | 108.94±0.87 | 63.34±1.02 | 74.31±0.99 | 16.82±0.47 | 14.46±0.33 | 44.36±0.32 |
| CIFAR10 | Mcdrop | 0.700±0.008 | 122.45±6.37 | 72.42±3.89 | 76.43±1.49 | 16.76±0.59 | 16.49±0.53 | 46.18±1.19 |
| (2500) | Aleatoric+MC | 0.704±0.022 | 119.76±15.06 | 70.66±7.69 | 75.29±2.35 | 16.55±1.56 | 16.30±1.23 | 45.62±3.49 |
| | CRL | 0.718±0.003 | 105.92±2.27 | 61.95±1.27 | 74.17±0.80 | 13.15±0.62 | 10.63±0.19 | 42.20±0.55 |
| | **Ours** | **0.755±0.004** | **81.50±2.70** | **48.72±1.75** | **71.42±1.08** | **5.77±0.30** | **8.56±0.24** | **35.24±0.53** |
| | Softmax | 0.795±0.002 | 60.54±1.45 | 38.02±1.14 | 68.64±1.08 | 14.73±0.20 | 11.26±0.15 | 34.28±0.47 |
| | AES | 0.799±0.002 | 51.68±1.04 | 29.99±0.80 | **60.44±0.90** | 8.16±0.19 | 9.46±0.09 | 28.51±0.21 |
| CIFAR10 | Mcdrop | 0.808±0.003 | 52.94±1.07 | 33.25±0.47 | 67.70±0.93 | 8.80±0.24 | 10.41±0.20 | 29.01±0.43 |
| (5000) | Aleatoric+MC | 0.810±0.001 | 50.96±1.47 | 31.73±1.36 | 67.46±1.22 | 8.67±0.24 | 10.22±0.16 | 28.69±0.30 |
| | CRL | 0.807±0.002 | 51.76±1.91 | 31.82±1.55 | 65.94±1.31 | 9.08±0.29 | 7.16±0.17 | 29.21±0.44 |
| | **Ours** | **0.818±0.002** | **44.43±1.03** | **26.88±0.66** | 64.64±0.93 | **5.26±0.43** | **6.42±0.08** | **26.88±0.43** |
| | Softmax | 0.849±0.001 | 37.46±1.09 | 25.44±0.93 | 63.34±1.88 | 11.03±0.13 | 8.49±0.09 | 25.59±0.26 |
| | AES | 0.855±0.004 | 30.75±0.89 | 19.71±0.57 | **57.09±0.71** | 6.61±0.43 | 7.23±0.21 | 21.68±0.57 |
| CIFAR10 | Mcdrop | 0.865±0.004 | 28.58±1.08 | 19.09±0.58 | 61.39±1.66 | 5.68±0.34 | 7.41±0.23 | **20.37±0.46** |
| (10000) | Aleatoric+MC | **0.865±0.001** | 28.64±0.70 | 19.22±0.65 | 61.80±2.13 | 5.75±0.20 | 7.47±0.10 | 20.47±0.23 |
| | CRL | 0.856±0.001 | 31.28±0.32 | 20.51±0.16 | 61.63±1.36 | 5.71±0.21 | 5.08±0.03 | 21.71±0.16 |
| | **Ours** | 0.860±0.002 | **28.39±0.54** | **18.19±0.36** | 59.23±1.96 | **3.99±0.17** | **4.83±0.03** | 20.77±0.18 |
| | | | | CIFAR100 | | | | |
| | Softmax | 0.292±0.006 | 506.70±4.52 | 159.23±6.38 | 79.15±1.54 | 31.89±1.34 | 38.25±0.99 | 97.18±0.96 |
| | AES | 0.289±0.009 | 509.38±12.07 | 157.50±3.01 | 79.93±1.38 | 30.17±0.83 | 38.63±0.79 | 95.77±0.39 |
| CIFAR100 | Mcdrop | 0.269±0.005 | 542.10±7.75 | 165.06±2.00 | 79.95±1.98 | 42.43±0.84 | 49.71±1.37 | 108.54±1.07 |
| (2500) | Aleatoric+MC | 0.269±0.005 | 542.86±10.72 | 165.56±3.95 | 80.96±1.63 | 42.48±1.70 | 49.61±2.02 | 108.81±1.78 |
| | CRL | 0.287±0.004 | 507.43±5.96 | 152.82±2.28 | 79.12±1.86 | 29.01±1.74 | 37.33±1.00 | 95.40±1.34 |
| | **Ours** | **0.365±0.004** | **399.49±6.23** | **133.15±3.78** | **75.36±0.82** | **27.55±0.35** | **33.54±0.39** | **87.24±0.66** |
| | Softmax | 0.425±0.005 | 344.09±5.31 | 132.95±0.95 | 77.10±1.18 | 32.74±0.55 | 35.28±0.47 | 86.17±0.73 |
| | AES | 0.419±0.005 | 335.02±4.86 | 118.75±0.87 | 73.30±0.73 | 22.89±0.69 | 34.04±0.63 | 77.93±0.74 |
| CIFAR100 | Mcdrop | 0.394±0.004 | 379.60±4.24 | 141.06±2.55 | 77.04±0.30 | 39.32±0.44 | 44.43±1.19 | 94.73±0.72 |
| (5000) | Aleatoric+MC | 0.394±0.002 | 378.72±3.47 | 140.54±1.48 | 77.48±0.80 | 39.12±0.49 | 44.19±0.95 | 94.67±0.73 |
| | CRL | 0.437±0.003 | 322.94±3.17 | 122.50±0.77 | 75.28±1.05 | 29.51±0.50 | 31.50±0.43 | 82.09±0.54 |
| | **Ours** | **0.482±0.005** | **266.08±5.87** | **100.22±2.40** | **71.50±1.88** | **18.61±0.54** | **23.99±0.47** | **70.33±0.91** |
| | Softmax | 0.546±0.004 | 214.17±2.81 | 90.78±0.59 | 72.72±0.52 | 24.31±0.40 | 24.19±0.44 | 67.41±0.59 |
| | AES | 0.546±0.002 | 209.77±3.62 | 86.38±2.24 | 71.55±1.12 | 19.63±0.20 | 24.29±0.46 | 63.62±0.42 |
| CIFAR100 | Mcdrop | 0.523±0.004 | 238.32±5.79 | 100.70±3.53 | 72.99±0.72 | 30.58±0.73 | 31.73±1.05 | 75.05±1.02 |
| (10000) | Aleatoric+MC | 0.521±0.006 | 240.02±6.93 | 100.94±3.05 | 73.15±1.49 | 30.54±0.55 | 31.79±1.16 | 75.27±1.22 |
| | CRL | 0.563±0.005 | 196.11±2.35 | 82.976±0.95 | 70.37±1.18 | 20.71±0.31 | 21.20±0.19 | 63.02±0.58 |
| | **Ours** | **0.590±0.003** | **168.57±2.39** | **70.26±0.89** | **68.30±1.37** | **14.34±0.32** | **17.55±0.22** | **56.23±0.49** |

## 4.2 MEDICAL IMAGE SEGMENTATION

Even though deep neural network has achieved very impressive performance in medical image segmentation (Litjens et al., 2017), it is still important to estimate the model confidence on samples, so the expert can get involved in time to avoid misjudgement. This emphasizes the importance of detecting and reacting to the failure of deep learning models. Thus, confidence estimation is a promising way to manage this reliability concern (Jungo & Reyes, 2019). On the other hand, image segmentation (dense label prediction) task is a good scenario to show the effectiveness of our consistency ranking loss, as the convergence of pairing loss is much more challenging. Here we use a publicly available dataset to conduct the experiment: the international skin imaging collaboration (ISIC) lesion segmentation dataset 2017 (Codella et al., 2018), which consists 2000 training, 150 validation and 600 testing annotated images.

**Implementation details.** We conduct experiments with various portions of labeled training samples: 1/16 (125), 1/8 (250), 1/4 (500) and full (2000). We use UNet (Ronneberger et al., 2015) with ResNet34 backbone as the model architecture for all methods. The loss weights $\lambda_1, \lambda_2$ are set to $0.5, 0.15$ for segmentation task. All models are trained by Adam with a learning rate 0.0001. We train our method for 200 epochs with a mini-batch size of 96 (32 labeled and 64 unlabeled). As for augmentation, a scheme with resizing ($192 \times 256$), random scale, random crop ($192 \times 256$), random horizontal and vertical flip is used. Here we use mean Intersection-Over-Union (IOU) to evaluate the segmentation quality. More details are provided in Append. A.1.

The results are shown in Tab. 4 (The results of 'full' setting are included in Append. A.4). We observe that our method can produce more reliable confidence estimates than other baselines. The lack of labeled training samples has little impact on the segmentation accuracy (mIOU), but deteriorates the confidence estimation quality severely. Our method shows robustness to the shortage of labeled samples in terms of confidence estimation, because of the effective utilization of unlabeled samples.

Table 3: The correctness supervision ablation study results on CIFAR10

| Dataset (labeled size) | Method | Acc↑ | AURC↓ | E-AURC↓ | FPR-95↓ | ECE↓ | NLL↓ | Brier↓ |
|---|---|---|---|---|---|---|---|---|
| | | | | CIFAR10 | | | | |
| 5000 | Point-wise (L1) | 0.803±0.004 | 76.73±5.79 | 55.88±4.87 | 66.97±0.75 | **3.25±0.71** | 7.84±0.64 | 28.51±0.71 |
| | Unified normalization | 0.812±0.001 | 48.03±2.31 | 29.07±2.26 | 64.83±0.79 | 9.34±0.36 | 7.33±0.06 | 29.87±0.02 |
| | w/o corr | 0.816±0.004 | 47.12±1.87 | 29.16±1.41 | 66.37±0.86 | 5.39±0.36 | 6.74±0.16 | 27.38±0.32 |
| | **ours** | **0.818±0.002** | **44.43±1.03** | **26.88±0.66** | **64.64±0.93** | 5.26±0.43 | **6.42±0.08** | **26.88±0.43** |

Table 4: Comparison of confidence estimates on ISIC2017. The value setting is the same as Tab. 2.

| Dataset (labeled size) | Method | mIOU↑ | AURC↓ | E-AURC↓ | FPR-95↓ | ECE↓ | NLL↓ | Brier↓ |
|---|---|---|---|---|---|---|---|---|
| | | | | ISIC2017 | | | | |
| 125 | Softmax | 0.801±0.005 | 34.36±7.76 | 31.33±7.92 | 60.90±2.11 | 6.45±0.30 | 3.82±0.46 | 13.96±0.21 |
| | AES | 0.802±0.006 | 21.12±1.10 | 18.07±1.01 | 57.26±4.65 | 5.46±0.37 | 4.17±0.22 | 12.84±0.53 |
| | Mcdrop | 0.801±0.005 | 30.23±3.81 | 27.19±3.75 | 61.45±1.25 | 6.36±0.30 | 4.05±0.33 | 13.88±0.49 |
| | Aleatoric+MC | 0.802±0.001 | 27.66±1.87 | 24.62±1.93 | 59.55±0.84 | 6.21±0.15 | 3.86±0.29 | 13.68±0.13 |
| | CRL | 0.810±0.004 | 22.14±2.82 | 19.33±2.81 | 62.15±2.75 | 4.11±0.54 | 2.86±0.34 | 12.25±0.42 |
| | **Ours** | **0.812±0.007** | **14.11±1.06** | **11.23±0.86** | **56.81±2.79** | **3.05±0.49** | **2.31±0.23** | **11.464±0.58** |
| 250 | Softmax | **0.819±0.002** | 28.91±4.70 | 26.45±4.72 | 58.51±0.93 | 5.62±0.20 | 3.35±0.21 | 12.45±0.18 |
| | AES | 0.819±0.002 | 18.08±1.38 | 15.60±1.43 | **54.09±2.26** | 4.57±0.21 | 3.41±0.09 | 11.38±0.16 |
| | Mcdrop | 0.819±0.008 | 25.38±4.76 | 22.89±4.57 | 58.31±1.49 | 5.45±0.40 | 3.39±0.23 | 12.27±0.67 |
| | Aleatoric+MC | 0.817±0.005 | 25.96±3.76 | 23.37±3.80 | 57.80±1.77 | 5.49±0.33 | 3.43±0.29 | 12.43±0.60 |
| | CRL | 0.818±0.002 | 16.43±0.94 | 13.90±0.90 | 62.56±9.11 | 3.93±0.91 | 2.60±0.25 | 11.64±0.82 |
| | **Ours** | 0.817±0.007 | **12.79±0.79** | **10.21±0.62** | 54.51±2.79 | **2.56±0.26** | **2.23±0.17** | **10.71±0.31** |
| 500 | Softmax | 0.822±0.003 | 23.45±0.93 | 21.09±0.90 | 56.90±1.49 | 5.22±0.22 | 2.98±0.17 | 11.88±0.33 |
| | AES | 0.823±0.006 | 15.74±1.18 | 13.39±1.13 | 54.29±3.51 | 4.39±0.15 | 3.10±0.14 | 11.10±0.24 |
| | Mcdrop | 0.821±0.003 | 21.21±2.57 | 18.83±2.55 | 56.69±1.04 | 4.61±0.27 | 2.68±0.17 | 11.49±0.27 |
| | Aleatoric+MC | 0.821±0.003 | 21.32±2.18 | 18.96±2.12 | 56.41±1.42 | 5.07±0.19 | 3.20±0.18 | 11.73±0.26 |
| | CRL | 0.822±0.006 | 14.04±1.03 | 11.64±1.00 | 55.46±2.30 | 2.82±0.25 | 2.33±0.15 | 10.64±0.16 |
| | **Ours** | **0.823±0.004** | **11.85±1.17** | **9.42±1.13** | **54.14±1.97** | **2.48±0.24** | **2.08±0.18** | **10.44±0.20** |

When it comes to confidence calibration, our method has superior performances under various sizes of labeled samples.

In Fig. 4, the qualitative results on ISIC2017 are showed, and we can observe that: 1) our method performs better for segmentation accuracy, 2) the maximum softmax output optimized on cross-entropy only is overconfident near the boundary of lesion region, 3) the confidence map generated by our method is more reasonable and consistent with training consistency.

**Ablation study for $\lambda_2$.** We also discuss the effect of weight changes on consistency ranking loss. As shown in Fig. 3, the perturbation of weight $\lambda_2$ has limited influence on confidence estimation performances, and our choice ($\lambda_2 = 0.15$) achieves the best outcome.

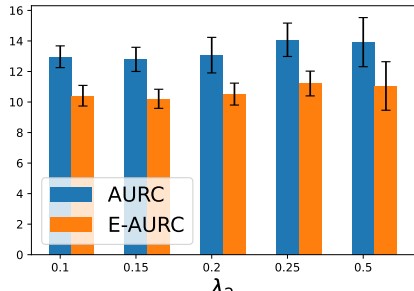

Figure 3: Ablation study for $\lambda_2$.

**With/without correctness supervision.** We used correctness ranking loss in our method to make full use of limited training labels. Here we conduct an ablation study to see the importance of the correctness ranking loss. According to Tab. 3 w/o corr, removing correctness ranking loss hurts the performance on both classification and segmentation tasks.

**Normalization strategy.** In this paper, we do min-max normalization for training consistency on labeled samples and unlabeled samples in mini-batch separately. Here we try to concatenate training consistency on labeled and unlabeled samples first then normalize mini-batch consistency together as an alternative solution. The results are shown in Tab. 3 Unified normalization. We can see all evaluation metrics deteriorate significantly with the alternative normalization strategy.

**Point-wise loss.** Here use a ranking loss to optimize confidence output, an alternative solution is to use a point-wise loss like L1 loss. However, due to training consistency is not of the same scale as the probability-based uncertainty, L1 loss cannot achieve good performance (Tab. 3).

## 4.3 ACTIVE LEARNING

In practice, there are scenarios where unlabeled samples are cheap to acquire but manual annotations are costly. To solve this problem, a promising direction is active learning, which requires learning algorithm actively queries experts for annotations. The critical issue here is how algorithm can

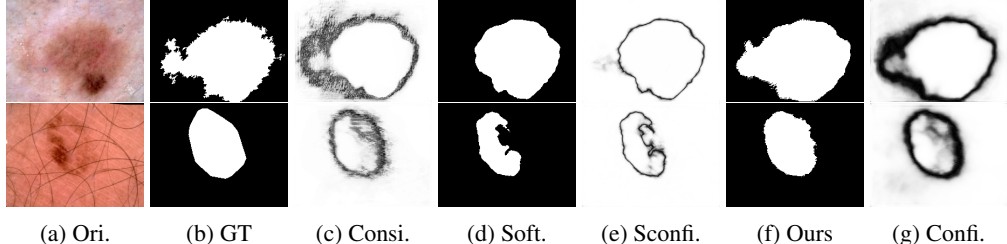

| (a) Ori. | (b) GT | (c) Consi. | (d) Soft. | (e) Sconfi. | (f) Ours | (g) Confi. |

Figure 4: Qualitative results on 250 labeled training samples compared to baselines. From left to right: **(a)** Original image, **(b)** GT label, **(c)** GT consistency map, **(d)** Segmentation map of cross entropy, **(e)** Maximum softmax of cross entropy (Softmax), **(f)** Segmentation map of proposed method, **(g)** Confidence map of the proposed method.

decide which unlabeled samples are valuable for the next stage learning. A common opinion is that the confidence levels of unlabeled samples denote the value for learning process. Therefore, in this section, we adopt the strategy, which queries the samples with least model certainty.

**Implementation details.** We evaluate active learning performance on CIFAR-10 and CIFAR-100. ResNet18 architecture is used for all models. For each stage, we train our model by SGD for 200 epochs with initial learning rate of 0.1, which shrinks a factor of 10 at 120 and 160 epochs. The rest hyper-parameters for our method are the same as Sec. 4.1. Here we compare our method with CRL, MC-dropout, core-set selection (Sener & Savarese, 2018), cross-entropy loss (using entropy as confidence output) and random sampling.

In the first training stage, 1000 labeled training samples and 49000 unlabeled samples are fed into training process. After training, 1000 samples from the unlabeled set with least model confidence are selected and annotated. We iteratively train and annotate for the next 9 stages. All 10 stages construct a complete training procedure. To make sure the experimental results are repeatable, we repeat each experiment for 5 times. The results are shown in Fig. 5. We can see our method performs

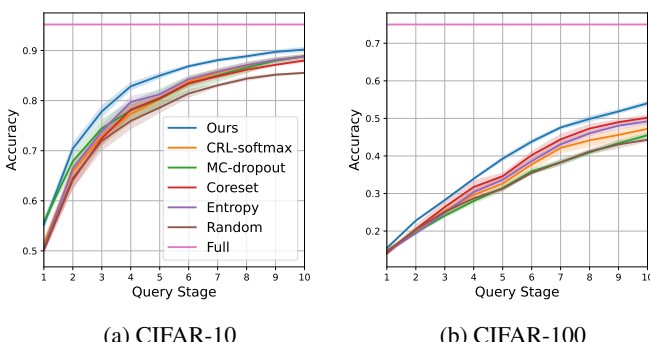

| (a) CIFAR-10 | (b) CIFAR-100 |

Figure 5: Active learning results comparison on CIFAR-10 and CIFAR-100. Curves are the mean accuracy of five iterations, shaded regions are standard deviations.

better in most stages for both CIFAR-10 and CIFAR-100. The reason is that our method can utilize the confidence information contained in unlabeled training samples by inspecting training consistency. This advantage is more apparent on CIFAR-100 because the shortage of training samples for each class is more severe for CIFAR-100. Through active learning experiment, we can conclude that the samples selected by confidence estimates generated by our method are more effective for the following active learning stages.

## 5 CONCLUSION

In this paper, we study the validation of training consistency in estimating the model confidence of unlabeled samples. Both t-SNE visualization and AURC (E-AURC) evaluation suggest that it generates reliable confidence estimates for both labeled and unlabeled samples with minor labeled samples. We propose consistency ranking loss to make the model confidence output be consistent with training consistency on labeled and unlabeled samples. We demonstrate the efficacy of consistency ranking loss by mathematical proof and empirical results. The superiority of our method on image classification and medical image segmentation tasks suggest wide application prospect of our method. The confidence estimates generated by our method also work for querying high quality training images and achieve better performance than other baselines in active learning.

**Reproducibility Statement:** We provide the necessary experimental details in Sec. 4. More specifically, the implementation details of image classification tasks are provided in Sec. 4.1. Sec. 4.2 contains the implementation details for image segmentation task. The implementation details of active learning tasks are provided in Sec. 4.3. The details of baseline methods are described in Append. A.1. The details of the pairing strategy are provided in Append. A.2. The details of evaluation metrics AURC and E-AURC are included in Sec. 3.2.

**Acknowledgement:** The authors thank anonymous reviewers for their constructive feedback. This research was partially supported by NSF CCF-2144901.

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

# A APPENDIX

Append. A.1 shows the implementation details of confidence estimation baselines.

Append. A.2 shows the details of mini-batch data pairing strategy.

Append. A.3 shows a detailed proof of Thm. 1

Append. A.4 shows the full confidence estimation results on ISIC2017

Append. A.5 shows the full confidence estimation results on CIFAR-10 and CIFAR-100

Append. A.6 shows the pseudo code of training model using our method.

Append. A.7 shows the anomaly detection results.

Append. A.8 shows the other possible ways for extracting consistency information.

Append. A.9 shows Cancer Survival dataset results.

## A.1 IMPLEMENTATION DETAILS OF CONFIDENCE ESTIMATION BASELINES

For image classification, to make sure all baselines are fully converged, We train baseline models for 2700, 1200, 600, 300 epochs on 1/20 (2500), 1/10 (5000), 1/5 (10000), 1 (50000) labeled training images. The mini-batch size is 128. The initial learning rate is 0.1 and shrinks a factor of 10 at $1/2 \times$ *number of epochs* and $5/6 \times$ *number of epochs*.

As for medical image segmentation (ISIC2017), we train all baseline models for 1500, 700, 300, 200 epochs on 1/16 (125), 1/8 (250), 1/4 (500), 1 (2000) labeled training images. The learning rate is 0.0001 all the time. The mini-batch size of all baselines is 64.

## A.2 DETAILS OF MINI-BATCH DATA PAIRING STRATEGY

The mini-batch data are constructed by labeled samples $D_l = \{(x_i, y_i)\}_{i=1}^b$ and unlabeled samples $D_u = \{(x_i^u, y_i^u)\}_{i=1}^d$. We define $x_{[b]} = x_0, x_{[i]} = x_{i+1}, i = 1, \ldots, b-1$ and $x_{[d]}^u = x_0^u, x_{[i]}^u = x_{i+1}^u, i = 1, \ldots, d$. In practice, we use empirical consistency ranking loss, which is defined by

$$\mathcal{L}_{con} = \sum_{s=1}^b \text{sign}(c_s - c_{[s]})((c_s - c_{[s]}) - (\kappa_s - \kappa_{[s]})) + \sum_{s=1}^d \text{sign}(c_s^u - c_{[s]}^u)((c_s^u - c_{[s]}^u) - (\kappa_s^u - \kappa_{[s]}^u))$$

where $c_s = c(x_s), c_{[s]} = c(x_{[s]}), \kappa_s = \kappa(x_s), \kappa_{[s]} = \kappa(x_{[s]}), c_s^u = c(x_s^u), c_{[s]}^u = c(x_{[s]}^u), \kappa_s^u = \kappa(x_s^u), \kappa_{[s]}^u = \kappa(x_{[s]}^u)$.

## A.3 DETAILED PROOF

Given $R_\kappa$, we define ranking set $H_\kappa = \{R_\kappa(x) : \hat{y} = y, \hat{y} = \arg\max_{i \in \mathcal{Y}} f(x)_i, (x, y) \in \mathcal{D}_m\}$. To make sure the duplicate value in $C_m$ won't affect the proof result, we stipulate that beside the rule above training consistency ranking function $R_c$ also need to minimize

$$\min_{\gamma \in \tau(H_\kappa, H_c)} \sum_{x \in T} |R_\kappa(x) - \gamma(R_\kappa(x))| \tag{7}$$

where $\tau$ is the collection of the bijective function between the elements of $H_\kappa$ and $H_c$. Satisfying two conditions above, we can define $R_c$ as an one-to-one function. Given $R_c$, we introduce a set $D_c = \{|c(R_c^{-1}(i)) - c(R_c^{-1}(j))|; i = \gamma_m(R_\kappa(x)), j = \gamma_m(R_\kappa(x)) + sign(R_\kappa(x) - \gamma_m(R_\kappa(x)))), x \in T, R_c^{-1}(j) \notin T(f|\mathcal{D}_m)\}$, where $\gamma_m = \arg\min_{\gamma \in \tau(H_\kappa, H_c)} \sum_{x \in T} |R_\kappa(x) - \gamma(R_\kappa(x))|$, $sign$ is a sign function and $c(x_i) = c_i, x_i \in X_m$.

To answer this question, we prove that consistency training loss is an upper bound of the absolute difference between the E-AURC of training consistency and confidence estimator on given dataset $\mathcal{D}_m = (X_m, Y_m)$ and training consistency $C_m$:

$$|AURC(f, c|\mathcal{D}_m) - AURC(f, \kappa|\mathcal{D}_m)| \leq \frac{1}{m \min D_c} \mathcal{L}_{cons}(f, X_m; C_m)$$

*Proof.* Let $T_\kappa(i)$ be the set constituted by all the correct predicted samples with confidence less or equal than $R_\kappa^{-1}(i)$, $T_\kappa(i) = \{x : R_\kappa(x) \leq i, \hat{y} = y, \hat{y} = \arg\max_{i \in \mathcal{Y}} f(x)_i, (x, y) \in \mathcal{D}_m\}$. $T_c(i)$ is the corresponding set introduced by training consistency $c$ and $T_c(i) = \{x : R_c(x) \leq i, \hat{y} = y, \hat{y} = \arg\max_{i \in \mathcal{Y}} f(x)_i, (x, y) \in \mathcal{D}_m\}$. We define $H_c(f, \kappa|\mathcal{D}_m) = \sum_{i=1}^m \mathbb{1}(\#T_\kappa(i) \neq \#T_c(i))$ and $H_c^k(f, \kappa|\mathcal{D}_m) = \sum_{i=1}^k \mathbb{1}(\#T_\kappa(i) \neq \#T_c(i))$. It can be proved that :

$$|AURC(f, c|\mathcal{D}_m) - AURC(f, \kappa|\mathcal{D}_m)| \leq \frac{H_c(f, \kappa|\mathcal{D}_m)}{m} \qquad (8)$$

There is a fact that $\forall i \in \{1, 2, \ldots, m\}$ if $\mathbb{1}(\#T_\kappa(i) = \#T_c(i))$, than $\hat{r}(f, g_{R_\kappa^{-1}(i)}|\mathcal{D}_m) = \hat{r}(f, g_{R_c^{-1}(i)}|\mathcal{D}_m)$, so

$$|AURC(f, c|\mathcal{D}_m) - AURC(f, \kappa|\mathcal{D}_m)| \leq \frac{1}{m} \sum_{\#T_\kappa(i) \neq \#T_c(i)} |\hat{r}(f, g_{R_\kappa^{-1}(i)}|\mathcal{D}_m) - \hat{r}(f, g_{R_c^{-1}(i)}|\mathcal{D}_m)|$$

$$\leq \frac{H_c(f, \kappa|\mathcal{D}_m)}{m}$$

We also have $F_r(f, c|\mathcal{D}_m) = \sum_{i=1}^m \prod_{j=1}^m \mathbb{1}(R_c(x_j) \leq i) \cdot \mathbb{1}(\hat{y}_j = y_j)$, where $\hat{y}_j = \arg\max_{i \in \mathcal{Y}} f(x_j)_i$ and $(x_j, y_j) \in \mathcal{D}_m$. Here we define $K(f, c|\mathcal{D}_m) = \#T(f|\mathcal{D}_m) - F_r(f, c|\mathcal{D}_m)$, which is the number of correct predicted samples with lower training consistency than at least one incorrect sample.

In the next part, we will prove that:

$$\frac{1}{m \min D_c} \mathcal{L}_{cons}(f, X_m; C_m) \geq \frac{H_c(f, \kappa|\mathcal{D}_m)}{m} \qquad (9)$$

We first discuss the case when $K(f, c|\mathcal{D}_m) = 0$. This means that the correct classified samples should have higher training consistency than the misclassified. Let $C_t$ be the set constructed by the consistency event count of correct classified samples, $C_t = \{c(x) : \hat{y} = y, \hat{y} = \arg\max_{i \in \mathcal{Y}} f(x)_i, (x, y) \in \mathcal{D}_m\}$ and $C_f$ be the set constructed by the consistency event count of false classified samples, $C_f = \{c(x) : \hat{y} \neq y, \hat{y} = \arg\max_{i \in \mathcal{Y}} f(x)_i, (x, y) \in \mathcal{D}_m\}$. It is not difficult to tell that $\min D_c = |\min C_t - \max C_f|$ under the case $K(f, c|\mathcal{D}_m) = 0$. Because $D_c = \{\min C_t - \max C_f\}$ when $K(f, c|\mathcal{D}_m) = 0$.

Here we have:

$$\frac{1}{|\min C_t - \max C_f|} \mathcal{L}_{cons}(f, X_m|C_m) = \sum_{s=1}^{m-1} \sum_{i=1, c_i < c_s}^m \frac{(|c_s - c_i| - (\kappa_s - \kappa_i))_+}{|\min C_t - \max C_f|}$$

$$\geq \sum_{s=1}^{m-1} \sum_{i=1}^m \mathbb{1}(l(\hat{y}_s, y_s) = 1 \cap l(\hat{y}_i, y_i) = 0 \cap \kappa_i > \kappa_s \cap c_i < c_s) \geq H_c(f, \kappa|\mathcal{D}_m)$$

The details of above formula please take a look at Theorem. 2. We should notice that the conclusion above does not rely on the data size of $\mathcal{D}_m$. Here we define the ranking function $R_T(\cdot, \kappa|\mathcal{D}_m)$ on $T(f|\mathcal{D}_m)$, $R_T(x, \kappa|\mathcal{D}_m) \in \{1, 2, \ldots, \#T(f|\mathcal{D}_m)\}, \forall x \in T(f|\mathcal{D}_m)$ and if $R_\kappa(x_i) < R_\kappa(x_j)$ than $R_T(x_i, \kappa|\mathcal{D}_m) < R_T(x_j, \kappa|\mathcal{D}_m), \forall x_i, x_j \in T(f|\mathcal{D}_m)$.

We assume the result stands when $K(f, c|\mathcal{D}_m) = n$. Than when $K(f, c|\mathcal{D}_m) = n + 1$, we start the discussion with the situation $R_T^{-1}(\#T(f|\mathcal{D}_m), \kappa|\mathcal{D}_m) = R_T^{-1}(\#T(f|\mathcal{D}_m), c|\mathcal{D}_m) = x_*$. According to assumption here, we can get that

$$\sum_{s=1}^{m-1} \sum_{i=1, c_i < c_s}^m \mathbb{1}\{x_i, x_s \neq x_*\} \frac{(|c_s - c_i| - (\kappa_s - \kappa_i))_+}{\min D_c} \geq H_c(f, \kappa|\mathcal{D}_m/\{x_*\})$$

There is a fact that $H_c^{\min\{R_\kappa(x_*), R_c(x_*)\}}(f, \kappa|\mathcal{D}_m/\{x_*\}) = H_c^{\min\{R_\kappa(x_*), R_c(x_*)\}}(f, \kappa|\mathcal{D}_m)$. Since $H_c(f, \kappa|\mathcal{D}_m/\{x_*\}) \geq H_c^{\min\{R_\kappa(x_*), R_c(x_*)\}}(f, \kappa|\mathcal{D}_m/\{x_*\})$ We have

$$\sum_{s=1}^{m-1} \sum_{i=1, c_i < c_s}^m \mathbb{1}\{x_i, x_s \neq x_*\} \frac{(|c_s - c_i| - (\kappa_s - \kappa_i))_+}{\min D_c} \geq H_c^{\min\{R_\kappa(x_*), R_c(x_*)\}}(f, \kappa|\mathcal{D}_m)$$

We begin below part with the scenarios $R_\kappa(x_*) \geq R_c(x_*)$, than it is straight that there must exist $|R_\kappa(x_*) - R_c(x_*)|$ samples from $F(f|\mathcal{D}_m)$ constructing a set $V = \{x \in X_m | R_\kappa(x) < R_\kappa(x_*), R_c(x) > R_c(x_*), x \in F(f|\mathcal{D}_m)\}$. It is easy to see that $\gamma_m(R_\kappa(x_*)) = R_c(x_*)$ and $sign(R_\kappa(x_*) - \gamma_m(R_\kappa(x_*))) = 1$. So $\forall x \in V, |c(x_*) - c(x)| \geq |c(x_*) - c(R_c^{-1}(R_c(x_*) + 1))| \in D_c$. Therefore, we can get that

$$\sum_{i=1, c_i \neq c_*}^{m} \frac{(|c_* - c_i| - (\kappa_* - \kappa_i))_+}{\min D_c} \geq \sum_{x_i \in V} \frac{(|c_* - c_i| - (\kappa_* - \kappa_i))_+}{\min D_c} \geq |R_\kappa(x_*) - R_c(x_*)|$$

As for $R_\kappa(x_*) < R_c(x_*)$, the proof is in the same manner. Summary the result above we get:

$$\frac{1}{\min D_c} \mathcal{L}_{cons}(f, X_m | C_m) = \sum_{s=1}^{m-1} \sum_{i=1, c_i < c_s}^{m} \mathbb{1}\{x_i, x_s \neq x_*\} \frac{(|c_s - c_i| - (\kappa_s - \kappa_i))_+}{\min D_c}$$
$$+ \sum_{i=1, c_i \neq c_*}^{m} \frac{(|c_* - c_i| - (\kappa_* - \kappa_i))_+}{\min D_c}$$
$$\geq H_c^{\min\{R_\kappa(x_*), R_c(x_*)\}}(f, \kappa|\mathcal{D}_m) + |R_\kappa(x_*) - R_c(x_*)| \geq H_c(f, \kappa|\mathcal{D}_m)$$

If $R_T^{-1}(\#T(f|\mathcal{D}_m), \kappa|\mathcal{D}_m) = x_1$, $R_T^{-1}(\#T(f|\mathcal{D}_m), c|\mathcal{D}_m) = x_2$ and $x_1 \neq x_2$, we can have one new confidence estimator by switching the confidence value between $x_1$ and $x_2$:

$$\kappa_t(x) = \begin{cases} \kappa(x_1) & x = x_2 \\ \kappa(x_2) & x = x_1 \\ \kappa(x) & otherwise \end{cases} \tag{10}$$

It is not difficult to see that consistency ranking loss introduced by $\kappa_t$: $\mathcal{L}_{cons}^t(f, X_m | \mathcal{D}_m)$ satisfying

$$\mathcal{L}_{cons}(f, X_m | C_m) \geq \mathcal{L}_{cons}^t(f, X_m | C_m)$$

To see this fact, we pay attention to single sample. First, we discuss sample $x_*$ satisfying $c(x_*) < c(x_2)$, we can define $a_l = c(x_2) - c(x_*) - (\kappa(x_2) - \kappa(x_*))$, $b_l = c(x_1) - c(x_*) - (\kappa(x_1) - \kappa(x_*))$, $a_t = c(x_2) - c(x_*) - (\kappa_t(x_2) - \kappa_t(x_*)) = c(x_2) - c(x_*) - (\kappa(x_1) - \kappa(x_*))$, $b_t = c(x_1) - c(x_*) - (\kappa_t(x_1) - \kappa_t(x_*)) = c(x_1) - c(x_*) - (\kappa(x_2) - \kappa(x_*))$. Apparently, $b_l > a_t, b_l > b_t, a_t > a_l, b_t > a_l$ and $a_l + b_l - (a_t + b_t) = 0$. Here we define function $g(x) = \max\{0, x\}$. It is straight that $g(a_l) + g(b_l) \geq g(a_t) + g(b_t)$. We discuss this by different situations: 1) if $b_l \leq 0$, then $g(a_l) + g(b_l) = g(a_t) + g(b_t) = 0$; 2) if $a_t \leq 0, b_t > 0$, since $b_l > b_t$, we have $g(a_l) + g(b_l) = g(b_l) = b_l > b_t = g(b_t) = g(a_t) + g(b_t)$; 3) if $b_l, a_t, b_t > 0, a_l < 0$, then $g(a_l) + g(b_l) \geq a_l + b_l \geq a_t + b_t = g(a_t) + g(b_t)$. As for other cases, they are easy to get.

Here we have $l(x_*) = g(a_l) + g(b_l)$ and $l_t(x_*) = g(a_t) + g(b_t)$, so $l(x_*) - l_t(x_*) \geq 0$.

As for $c(x_1) \geq c(x_*) \geq c(x_2)$, we have $a_l = c(x_1) - c(x_*) - (\kappa(x_1) - \kappa(x_*))$, $b_l = c(x_*) - c(x_2) - (\kappa(x_*) - \kappa(x_2))$, $a_t = c(x_1) - c(x_*) - (\kappa(x_2) - \kappa(x_*))$, $b_t = c(x_*) - c(x_2) - (\kappa(x_*) - \kappa(x_1))$. Apparently $a_l \geq a_t$ and $b_l \geq b_t$, so we have $g(a_l) + g(b_l) \geq g(a_t) + g(b_t)$. Also $l(x_*) \geq l_t(x_*)$.

As for $c(x_*) > c(x_1)$, the proof is the same as the situation $c(x_*) < c(x_2)$.

If $x_* = x_1$, than $l(x_*) = c(x_1) - c(x_2) - (\kappa(x_1) - \kappa(x_2))$ and $l_t(x_*) = c(x_1) - c(x_2) - (\kappa(x_2) - \kappa(x_1))$. It is straight that $l(x_*) \geq l_t(x_*)$.

Summary the result above, we can know that $\mathcal{L}_{cons}(f, X_m | C_m) - \mathcal{L}_{cons}^t(f, X_m | C_m) = \sum_{x \in X_m, x \neq x_2} l(x) - l_t(x) \geq 0$

So the proof stands for case $K(f, c|\mathcal{D}_m) = n + 1$, under the assumption it is true while $K(f, c|\mathcal{D}_m) = n$. $\square$

Let $T_s$ be the set constituted by all the correct predicted samples, $T_s = \{x : l(\hat{y}, y) = 0, \hat{y} = \arg\max_{i \in \mathcal{Y}} f(x)_i, (x, y) \in \mathcal{D}_m\}$. $T_{rank} = \{R_\kappa(t) : t \in T_s\}$ is the corresponding ranking set. We also define $F_s$ to be the wrong predicted set, which is $F_s = \{x : l(\hat{y}, y) = 1, \hat{y} = \arg\max_{i \in \mathcal{Y}} f(x)_i, (x, y) \in \mathcal{D}_m\}$ and $F_{rank} = \{R_\kappa(d) : d \in F_s\}$.

**Theorem 2.** *Given dataset with $m$ samples $D_m$, training consistency $C_m$, confidence estimator $\kappa$ and classifier $f$, we have*

$$\frac{H_c(f, \kappa | \mathcal{D}_m)}{m} \leq \frac{1}{m |\min C_t - \max C_f|} \mathcal{L}_{cons}(f, X_m; C_m)$$

*Proof.* In this proof, we don't consider the situation when $f$ is an optimal classifier or $\kappa$ is a perfect confidence estimator (correctly predicted samples have higher confidence that the wrong ones). Because for those two cases, The E-AURC difference will always be 0. Here we define $F_r(f, \kappa | \mathcal{D}_m) = \min F_{rank}$, $T_r(f, \kappa | \mathcal{D}_m) = \max T_{rank}$. It is straight that $H_c(f, \kappa | \mathcal{D}_m) = T_r(f, \kappa | \mathcal{D}_m) - F_r(f, \kappa | \mathcal{D}_m)$ when $K(f, c | \mathcal{D}_m) = 0$.

First we show that $\forall x_i, x_j \in X_m$, if $\kappa_i > \kappa_s$, $c_i < c_s$, $l(\hat{y}_s, y_s) = 1$ and $l(\hat{y}_i, y_i) = 0$, than $|c_s - c_i| \geq (\min C_t - \max C_f)$. It is apparent that $c_s \in C_f$ and $c_i \in C_t$, so we have $c_s \geq \min C_f$ and $c_i \leq \max C_t$. This inequation thus stands. So far we have

$\frac{1}{|\min C_t - \max C_f|} \mathcal{L}_{cons}(f, X_m | C_m) \quad = \quad \sum_{s=1}^{n-1} \sum_{i=1, c_i < c_s}^{n} \frac{(|c_s - c_i| - (\kappa_s - \kappa_i))_+}{|\min C_t - \max C_f|} \quad \geq$
$\sum_{s=1}^{n-1} \sum_{i=1, c_i < c_s}^{n} \mathbb{1}(l(\hat{y}_s, y_s) = 1 \cap l(\hat{y}_i, y_i) = 0 \cap \kappa_i > \kappa_s) \frac{(|c_s - c_i| - (\kappa_s - \kappa_i))_+}{|\min C_t - \max C_f|} \quad \geq$
$\sum_{s=1}^{n-1} \sum_{i=1}^{n} \mathbb{1}(l(\hat{y}_s, y_s) = 1 \cap l(\hat{y}_i, y_i) = 0 \cap \kappa_i > \kappa_s \cap c_i < c_s) = I(f, \kappa, X_m | \mathcal{D}_m)$

Next, we prove that for any confidence score function $\kappa$, there is a fact that

$$I(f, \kappa, X_m | \mathcal{D}_m) = I(f, X_m | \mathcal{D}_m) \geq T_r(f, \kappa | \mathcal{D}_m) - F_r(f, \kappa | \mathcal{D}_m)$$

Here we define $K_m = \{x \in X_m : R_\kappa(x) > F_r(f, \kappa | \mathcal{D}_m), l(\hat{y}, y) = 0\}$. If $\#K_m = 0$, than $\forall x_i, x_j \in X_m$, $l(\hat{y}_i, y_i) = 1, l(\hat{y}_j, y_j) = 0$, we have $R_\kappa(x_j) < R_\kappa(x_i)$. This property makes $\kappa$ under this condition an optimal confidence estimator and $T_r(f, \kappa | \mathcal{D}_m) < F_r(f, \kappa | \mathcal{D}_m)$. It is straight that $I(f, X_m | \mathcal{D}_m) \geq 0$. Therefore, we have $I(f, X_m | \mathcal{D}_m) \geq T_r(f, \kappa | \mathcal{D}_m) - F_r(f, \kappa | \mathcal{D}_m)$

If $\#K_m = 1$, we have $K_m = \{x_*\}$, $(x_*, y_*) \in \mathcal{D}_m$ and $G_m = \{x \in X_m : R_\kappa(x) < T_r(f, \kappa | \mathcal{D}_m), l(\hat{y}, y) = 1\}$. Since $l(\hat{y}_*, y_*) = 0$, it is clear that $R_\kappa(x_*) \leq T_r(f, \kappa | \mathcal{D}_m)$. But if $R_\kappa(x_*) < T_r(f, \kappa | \mathcal{D}_m)$, we can get $x_t = R_\kappa^{-1}(T_r(f, \kappa | \mathcal{D}_m))$. Apparently $x_t \in T_s$ and $R_\kappa(x_t) > R_\kappa(x_*) > F_r(f, \kappa | \mathcal{D}_m)$, so $x_t \in K_m$, which conflicts the fact that $\#K_m = 1$. We have $R_\kappa(x_*) = T_r(f, \kappa | \mathcal{D}_m)$. According to our assumption about training consistency, it is apparent that $\forall x_\# \in G_m, (x_\#, y_\#) \in \mathcal{D}_m, l(\hat{y}_*, y_*) = 0, l(\hat{y}_\#, y_\#) = 1, \kappa(x_*) < \kappa(x_\#)$ and $C(x_*) > C(x_\#)$, where $\hat{y}_* = \arg\max_{i \in \mathcal{Y}} f(x_*)_i$ and $\hat{y}_\# = \arg\max_{i \in \mathcal{Y}} f(x_\#)_i$. Here we also have $\#G_m \geq T_r(f, \kappa | \mathcal{D}_m) - F_r(f, \kappa | \mathcal{D}_m)$. To prove this, we assume $\#G_m < T_r(f, \kappa | \mathcal{D}_m) - F_r(f, \kappa | \mathcal{D}_m)$. We have if $x \in G_m$, than $R_\kappa(x) \geq F_r(f, \kappa | \mathcal{D}_m)$, because $F_r(f, \kappa | \mathcal{D}_m)$ is the minimum value of $F_{rank}$. Therefore, if $\#G_m < T_r(f, \kappa | \mathcal{D}_m) - F_r(f, \kappa | \mathcal{D}_m)$, there must exist $x_k \in X_m$ such that $F_r(f, \kappa | \mathcal{D}_m) < R_\kappa(x_k) < T_r(f, \kappa | \mathcal{D}_m)$ and $x_k \in T_s$, which makes $x_k \in K_m$. Apparently $x_k$ and $x_*$ are two different elements of $K_m$, which conflicts the fact that $\#K_m = 1$. This is a simple fact that $I(f, X_m \mathcal{D}_m) \geq \sum_{x_\# \in G_m} \mathbb{1}(x_* \in T \cap x_\# \in F \cap \kappa(x_*) < \kappa(x_\#) \cap C(x_*) > C(x_\#)) \geq \sum_{x_\# \in G_m} 1 \geq T_r(f, \kappa | \mathcal{D}_m) - F_r(f, \kappa | \mathcal{D}_m)$.

In the context below, we show that for every $k \geq 1$, if $\#K_m = k$ this inequation holds, than when $\#K_m = k + 1$, this inequation also holds. Otherwise, we assume this inequation holds for any confidence estimator satisfying $\#K_m = k$, but there exists confidence estimator $\kappa_f$ such that $I(f, \kappa_f, X_m | \mathcal{D}_m) < T_r(f, \kappa_f | \mathcal{D}_m) - F_r(f, \kappa_f | \mathcal{D}_m), K_m^{\kappa_f} = \{x \in X_m : R_{\kappa_f}(x) > F_r(f, \kappa_f | \mathcal{D}_m), l(\hat{y}, y) = 0\}$ and $\#K_m^{\kappa_f} = k + 1$. Here we can have a confidence estimator

$$\kappa_t(x) = \begin{cases} \kappa_f(R_{\kappa_f}^{-1}(F(f, \kappa_f | \mathcal{D}_m))) & x = \arg\min_{x \in K_m^{\kappa_f}} R_{\kappa_f}(x) \\ \kappa_f(\arg\min_{x \in K_m^{\kappa_f}} R_{\kappa_f}(x)) & x = R_{\kappa_f}^{-1}(F(f, \kappa_f | \mathcal{D}_m)) \\ \kappa_f(x) & otherwise \end{cases}$$

It is apparent that we get $\kappa_t$ by swapping the confidence score of $\kappa_f$ at $\arg\min_{x \in K_m^{\kappa_f}} R_{\kappa_f}(x)$ and $R_{\kappa_f}^{-1}(F(f, \kappa_f | \mathcal{D}_m))$, so $\#K_m^{\kappa_t} = k$ and $I(f, \kappa_t, X_m | \mathcal{D}_m) \geq T_r(f, \kappa_t | \mathcal{D}_m) - F_r(f, \kappa_t | \mathcal{D}_m)$. It is also straight to see that $T_r(f, \kappa_t | \mathcal{D}_m) = T_r(f, \kappa_f | \mathcal{D}_m)$, $F_r(f, \kappa_f | \mathcal{D}_m) = F_r(f, \kappa_t | \mathcal{D}_m) - 1$. We also get that $I(f, \kappa_f, X_m | \mathcal{D}_m) - I(f, \kappa_t, X_m | \mathcal{D}_m) \geq 1$, because

Table 5: Comparison of confidence estimates on ISIC2017. The value setting is the same as Tab. 2

| Dataset (labeled size) | Method | mIOU↑ | AURC↓ | E-AURC↓ | FPR-95↓ | ECE↓ | NLL↓ | Brier↓ |
|---|---|---|---|---|---|---|---|---|
| | | | | ISIC2017 | | | | |
| 125 | Softmax | 0.801±0.005 | 34.36±7.76 | 31.33±7.92 | 60.90±2.11 | 6.45±0.30 | 3.82±0.46 | 13.96±0.21 |
| | AES | 0.802±0.006 | 21.12±1.10 | 18.07±1.01 | 57.26±4.65 | 5.46±0.37 | 4.17±0.22 | 12.84±0.53 |
| | Mcdrop | 0.801±0.005 | 30.23±3.81 | 27.19±3.75 | 61.45±1.25 | 6.36±0.30 | 4.05±0.33 | 13.88±0.49 |
| | Aleatoric+MC | 0.802±0.001 | 27.66±1.87 | 24.62±1.93 | 59.55±0.84 | 6.21±0.15 | 3.86±0.29 | 13.68±0.13 |
| | CRL | 0.810±0.004 | 22.14±2.82 | 19.33±2.81 | 62.15±2.75 | 4.11±0.54 | 2.86±0.34 | 12.25±0.42 |
| | **Ours** | **0.812±0.007** | **14.11±1.06** | **11.23±0.86** | **56.81±2.79** | **3.05±0.49** | **2.31±0.23** | **11.464±0.58** |
| 250 | Softmax | **0.819±0.002** | 28.91±4.70 | 26.45±4.72 | 58.51±0.93 | 5.62±0.20 | 3.35±0.21 | 12.45±0.18 |
| | AES | 0.819±0.002 | 18.08±1.38 | 15.60±1.43 | **54.09±2.26** | 4.57±0.21 | 3.41±0.09 | 11.38±0.16 |
| | Mcdrop | 0.819±0.008 | 25.38±4.76 | 22.89±4.57 | 58.31±1.49 | 5.45±0.40 | 3.39±0.23 | 12.27±0.67 |
| | Aleatoric+MC | 0.817±0.005 | 25.96±3.76 | 23.37±3.80 | 57.80±1.77 | 5.49±0.33 | 3.43±0.29 | 12.43±0.60 |
| | CRL | 0.818±0.002 | 16.43±0.94 | 13.90±0.90 | 62.56±9.11 | 3.93±0.91 | 2.60±0.25 | 11.64±0.82 |
| | **Ours** | 0.817±0.007 | **12.79±0.79** | **10.21±0.62** | 54.51±2.79 | **2.56±0.26** | **2.23±0.17** | **10.71±0.31** |
| 500 | Softmax | 0.822±0.003 | 23.45±0.93 | 21.09±0.90 | 56.90±1.49 | 5.22±0.22 | 2.98±0.17 | 11.88±0.33 |
| | AES | **0.823±0.006** | 15.74±1.18 | 13.39±1.13 | 54.29±3.51 | 4.39±0.15 | 3.10±0.14 | 11.10±0.24 |
| | Mcdrop | 0.821±0.003 | 21.21±2.57 | 18.83±2.55 | 56.69±1.04 | 4.61±0.27 | 2.68±0.17 | 11.49±0.27 |
| | Aleatoric+MC | 0.821±0.003 | 21.32±2.18 | 18.96±2.12 | 56.41±1.42 | 5.07±0.19 | 3.20±0.18 | 11.73±0.26 |
| | CRL | 0.822±0.006 | 14.04±1.03 | 11.64±1.00 | 55.46±2.30 | 2.82±0.25 | 2.33±0.15 | 10.64±0.16 |
| | **Ours** | 0.823±0.004 | **11.85±1.17** | **9.42±1.13** | **54.14±1.97** | **2.48±0.24** | **2.08±0.18** | **10.44±0.20** |
| full:2000 | Softmax | 0.831±0.003 | 16.39±0.61 | 14.29±0.61 | 53.69±1.00 | 4.67±0.19 | 2.87±0.17 | 10.94±0.25 |
| | AES | 0.829±0.004 | 12.54±0.61 | 10.40±0.60 | **51.02±2.32** | 3.98±0.13 | 3.09±0.07 | 10.40±0.14 |
| | Mcdrop | 0.834±0.005 | 15.41±0.92 | 13.40±0.89 | 54.32±0.49 | 4.51±0.18 | 2.86±0.08 | 10.69±0.33 |
| | Aleatoric+MC | 0.828±0.010 | 14.81±1.05 | 12.66±1.07 | 54.37±0.65 | 4.57±0.32 | 3.04±0.21 | 10.94±0.58 |
| | CRL | 0.834±0.001 | **10.17±0.46** | **8.08±0.46** | 51.83±3.18 | 2.61±0.27 | 2.02±0.07 | 9.76±0.21 |
| | **Ours** | **0.838±0.007** | 11.18±1.39 | 9.17±1.35 | 52.27±3.14 | **2.23±0.43** | **1.89±0.17** | **9.49±0.43** |

$I(f, \kappa_f, X_m | \mathcal{D}_m) - I(f, \kappa_t, X_m | \mathcal{D}_m) = \sum_{x: F(f, \kappa_f | \mathcal{D}_m) \leq R_{\kappa_f}(x) < \arg\min_{x \in K_m^{\kappa_f}} R_{\kappa_f}(x)} \mathbb{1}(x \in F, x_1 = R_{\kappa_f}^{-1}(F(f, \kappa_f | \mathcal{D}_m)) \in T, \kappa(x) > \kappa(x_1), C(x) < C(x_1)) \geq 1$. So far according to our assumption we have

$$I(f, \kappa_f, X_m | \mathcal{D}_m) < T_r(f, \kappa_f | \mathcal{D}_m) - F_r(f, \kappa_f | \mathcal{D}_m) \implies$$
$$I(f, \kappa_t, X_m | \mathcal{D}_m) + 1 < T_r(f, \kappa_f | \mathcal{D}_m) - F_r(f, \kappa_f | \mathcal{D}_m) \implies$$
$$I(f, \kappa_t, X_m | \mathcal{D}_m) + 1 < T_r(f, \kappa_t | \mathcal{D}_m) - F_r(f, \kappa_t | \mathcal{D}_m) + 1 \implies$$
$$I(f, \kappa_t, X_m | \mathcal{D}_m) < T_r(f, \kappa_t | \mathcal{D}_m) - F_r(f, \kappa_t | \mathcal{D}_m)$$

Which leads to a conflict.

$\square$

### A.4 ISIC2017 CONFIDENCE ESTIMATION RESULTS IN TAB. 5

### A.5 CIFAR-10 AND CIFAR-100 CONFIDENCE ESTIMATION RESULTS IN TAB. 6

### A.6 PSEUDO CODE

Here we provide the pseudo code 1 for the training process of consistency ranking loss. Here we need to notice that we normalize the training consistency for labeled samples and unlabeled samples in a mini-batch independently. The reason is that training consistency has a bias on labeled training samples, so the labeled samples have higher training consistency than unlabeled ones. There is a mismatch between the training consistency distribution of labeled samples and unlabeled samples. A simple way to overcome this issue is Min-max normalize the consistency of labeled and unlabeled samples separately. In practice, this will achieve better performance.

### A.7 ANOMALY DETECTION

We evaluate the anomaly detection ability of our method on CIFAR10 and use the samples of CIFAR100 as out-of-distribution samples. In this part, we use the 10% (5000) labeled samples setting. The results are shown in the table below, and our method achieves best performance. This indicates that the deep classifier trained by our method can detect out-of-distribution samples efficiently. For the training details please refer to Sec. 4.1. We compare our method with Cross-entropy loss and the correctness ranking loss. Here we use detection error (the minimum error among all thresh-

Table 6: Comparison of confidence estimates on CIFAR-10/100 with various labeled training data size. The best results of experiments are shown in bold. The values of AURC & E-AURC are multiplied by $10^3$, FPR are multiplied by $10^2$ and NLL is multiplied by 10 for clarity

| Dataset (labeled size) | Method | Acc↑ | AURC↓ | E-AURC↓ | FPR-95↓ | ECE↓ | NLL↓ | Brier↓ |
|---|---|---|---|---|---|---|---|---|
| | | | | CIFAR10 | | | | |
| CIFAR10 (2500) | Softmax | 0.717±0.008 | 109.78±6.69 | 65.32±3.92 | 75.64±0.59 | 20.23±0.76 | 14.83±0.60 | 47.17±1.53 |
| | AES | 0.713±0.002 | 108.94±0.87 | 63.34±1.02 | 74.31±0.99 | 16.82±0.47 | 14.46±0.33 | 44.36±0.32 |
| | Mcdrop | 0.700±0.008 | 122.45±6.37 | 72.42±3.89 | 76.43±1.49 | 16.76±0.59 | 16.49±0.53 | 46.18±1.19 |
| | Aleatoric+MC | 0.704±0.022 | 119.76±15.06 | 70.66±7.69 | 75.29±2.35 | 16.55±1.56 | 16.30±1.23 | 45.62±3.49 |
| | CRL | 0.718±0.003 | 105.92±2.27 | 61.95±1.27 | 74.17±0.80 | 13.15±0.62 | 10.63±0.19 | 42.20±0.55 |
| | **Ours** | **0.755±0.004** | **81.50±2.70** | **48.72±1.75** | **71.42±1.08** | **5.77±0.30** | **8.56±0.24** | **35.24±0.53** |
| CIFAR10 (5000) | Softmax | 0.795±0.002 | 60.54±1.45 | 38.02±1.14 | 68.64±1.08 | 14.73±0.20 | 11.26±0.15 | 34.28±0.47 |
| | AES | 0.799±0.002 | 51.68±1.04 | 29.99±0.80 | **60.44±0.90** | 8.16±0.19 | 9.46±0.09 | 28.51±0.21 |
| | Mcdrop | 0.808±0.003 | 52.94±1.07 | 33.25±0.47 | 67.70±0.93 | 8.80±0.24 | 10.41±0.20 | 29.01±0.43 |
| | Aleatoric+MC | 0.810±0.001 | 50.96±1.47 | 31.73±1.36 | 67.46±1.22 | 8.67±0.24 | 10.22±0.16 | 28.69±0.30 |
| | CRL | 0.807±0.002 | 51.76±1.91 | 31.82±1.55 | 65.94±1.31 | 9.08±0.29 | 7.16±0.17 | 29.21±0.44 |
| | **Ours** | **0.818±0.002** | **44.43±1.03** | **26.88±0.66** | 64.64±0.93 | **5.26±0.43** | **6.42±0.08** | **26.88±0.43** |
| CIFAR10 (10000) | Softmax | 0.849±0.001 | 37.46±1.09 | 25.44±0.93 | 63.34±1.88 | 11.03±0.13 | 8.49±0.09 | 25.59±0.26 |
| | AES | 0.855±0.004 | 30.75±0.89 | 19.71±0.57 | **57.09±0.71** | 6.61±0.43 | 7.23±0.21 | 21.68±0.57 |
| | Mcdrop | 0.865±0.004 | 28.58±1.08 | 19.09±0.58 | 61.39±1.66 | 5.68±0.34 | 7.41±0.23 | **20.37±0.46** |
| | Aleatoric+MC | **0.865±0.001** | 28.64±0.70 | 19.22±0.65 | 61.80±2.13 | 5.75±0.20 | 7.47±0.10 | 20.47±0.23 |
| | CRL | 0.856±0.001 | 31.28±0.32 | 20.51±0.16 | 61.63±1.36 | 5.71±0.21 | 5.08±0.03 | 21.71±0.16 |
| | **Ours** | 0.860±0.002 | **28.39±0.54** | **18.19±0.36** | 59.23±1.96 | **3.99±0.17** | **4.83±0.03** | 20.77±0.18 |
| CIFAR10 (Full: 50000) | Softmax | 0.941±0.002 | 9.11±0.44 | 7.34±0.39 | 40.42±2.30 | 4.46±0.16 | 3.34±0.13 | 10.19±0.32 |
| | AES | 0.942±0.002 | 5.80±0.28 | 4.09±0.25 | **36.37±2.85** | 1.61±0.20 | 1.82±0.04 | 8.69±0.29 |
| | Mcdrop | 0.942±0.000 | **5.48±0.19** | **3.80±0.16** | 36.74±3.06 | 1.45±0.15 | 1.88±0.05 | 8.48±0.13 |
| | Aleatoric+MC | **0.943±0.000** | 6.02±0.33 | 4.38±0.30 | 38.72±1.82 | 1.25±0.07 | 1.80±0.03 | **8.36±0.12** |
| | CRL | 0.940±0.001 | 6.02±0.26 | 4.21±0.19 | 38.81±1.59 | 1.23±0.18 | 1.81±0.04 | 8.85±0.20 |
| | **Ours** | 0.942±0.001 | 5.83±0.25 | 4.16±0.16 | 40.69±1.37 | **0.86±0.07** | **1.76±0.02** | 8.60±0.16 |
| | | | | CIFAR100 | | | | |
| CIFAR100 (2500) | Softmax | 0.292±0.006 | 506.70±4.52 | 159.23±6.38 | 79.15±1.54 | 31.89±1.34 | 38.25±0.99 | 97.18±0.96 |
| | AES | 0.289±0.009 | 509.38±12.07 | 157.50±3.01 | 79.93±1.38 | 30.17±0.83 | 38.63±0.79 | 95.77±0.39 |
| | Mcdrop | 0.269±0.005 | 542.10±7.75 | 165.06±2.00 | 79.95±1.98 | 42.43±0.84 | 49.71±1.37 | 108.54±1.07 |
| | Aleatoric+MC | 0.269±0.005 | 542.86±10.72 | 165.56±3.95 | 80.96±1.63 | 42.48±1.70 | 49.61±2.02 | 108.81±1.78 |
| | CRL | 0.287±0.004 | 507.43±5.96 | 152.82±2.28 | 79.12±1.86 | 29.01±1.74 | 37.33±1.00 | 95.40±1.34 |
| | **Ours** | **0.365±0.004** | **399.49±6.23** | **133.15±3.78** | **75.36±0.82** | **27.55±0.35** | **33.54±0.39** | **87.24±0.66** |
| CIFAR100 (5000) | Softmax | 0.425±0.005 | 344.09±5.31 | 132.95±0.95 | 77.10±1.18 | 32.74±0.55 | 35.28±0.47 | 86.17±0.73 |
| | AES | 0.419±0.005 | 335.02±4.86 | 118.75±0.87 | 73.30±0.73 | 22.89±0.69 | 34.04±0.63 | 77.93±0.74 |
| | Mcdrop | 0.394±0.004 | 379.60±4.24 | 141.06±2.55 | 77.04±0.30 | 39.32±0.44 | 44.43±1.19 | 94.73±0.72 |
| | Aleatoric+MC | 0.394±0.002 | 378.72±3.47 | 140.54±1.48 | 77.48±0.80 | 39.12±0.49 | 44.19±0.95 | 94.67±0.73 |
| | CRL | 0.437±0.003 | 322.94±3.17 | 122.50±0.77 | 75.28±1.05 | 29.51±0.50 | 31.50±0.43 | 82.09±0.54 |
| | **Ours** | **0.482±0.005** | **266.08±5.87** | **100.22±2.40** | **71.50±1.88** | **18.61±0.54** | **23.99±0.47** | **70.33±0.91** |
| CIFAR100 (10000) | Softmax | 0.546±0.004 | 214.17±2.81 | 90.78±0.59 | 72.72±0.52 | 24.31±0.40 | 24.19±0.44 | 67.41±0.59 |
| | AES | 0.546±0.002 | 209.77±3.63 | 86.38±2.24 | 71.55±1.12 | 19.63±0.20 | 24.29±0.46 | 63.62±0.42 |
| | Mcdrop | 0.523±0.004 | 238.32±5.79 | 100.70±3.53 | 72.99±0.72 | 30.58±0.73 | 31.73±1.05 | 75.05±1.02 |
| | Aleatoric+MC | 0.521±0.006 | 240.02±6.93 | 100.94±3.05 | 73.15±1.49 | 30.54±0.55 | 31.79±1.16 | 75.27±1.22 |
| | CRL | 0.563±0.005 | 196.11±2.35 | 82.976±0.95 | 70.37±1.18 | 20.71±0.31 | 21.20±0.19 | 63.02±0.58 |
| | **Ours** | **0.590±0.003** | **168.57±2.39** | **70.26±0.89** | **68.30±1.37** | **14.34±0.32** | **17.55±0.22** | **56.23±0.49** |
| CIFAR100 (Full: 50000) | Softmax | 0.753±0.002 | 71.75±0.89 | 38.63±0.72 | 63.30±1.93 | 12.67±0.25 | 11.54±0.08 | 37.26±0.21 |
| | AES | 0.760±0.001 | 65.22±0.73 | 33.95±0.68 | 62.17±0.54 | 7.38±0.22 | 9.04±0.04 | 33.96±0.16 |
| | Mcdrop | 0.758±0.003 | 66.92±1.45 | 34.97±0.46 | 63.27±1.47 | **5.59±0.33** | 9.49±0.14 | 34.02±0.38 |
| | Aleatoric+MC | 0.755±0.003 | 67.87±1.55 | 35.05±0.65 | 61.69±1.79 | 6.01±0.22 | 9.45±0.13 | 34.25±0.47 |
| | CRL | 0.768±0.002 | 61.77±1.07 | 32.57±0.81 | 61.79±2.20 | 8.59±0.17 | 9.11±0.09 | 33.39±0.28 |
| | **ours** | **0.780±0.002** | **55.40±0.90** | **29.36±0.57** | **60.67±1.03** | 7.87±0.18 | **8.45±0.07** | **31.52±0.35** |

olds) and the area under the receiver operating characteristic curve (AUROC) to evaluate the out-of-distribution performances. All experiments are iterated 3 times and shown in Tab. 7. The mean and standard deviation are reported.

## A.8 ADDITIONAL QUALITATIVE RESULTS

In this paper, we use consistency events to quantify the sensitive levels of samples vibrating around the decision boundary in the training process. It's worth mentioning that this is not the only way to capture such information. For example, we define label frequency $r_i^y = \sum_{t=0}^{T-1} \mathbb{1}\{\hat{y}_i^t = y\}$ as a variable to count the number of sample $x_i$ being predicted as label $y$ by classifier during training and it is also able to capture training consistency. We can use the maximum label frequency (L-count) $\max_{y \in \mathcal{Y}} r_i^y$ to quantify the training consistency on sample $x_i$. Apparently, the higher maximum label frequency introduces less vacillation in the training process, leading to higher confidence. Besides, we can also use the label frequency entropy (L-entropy) and margin (L-margin) to summarize

---

**Algorithm 1:** Consistency ranking loss training

---

**Input:** Dataloader for labeled and labeled training samples
**Output:** Trained deep model
**Definition**: $uLoader$ and $sLoader$ denote the dataloader for unlabeled and labeled samples; $D_{corr}$ is the dictionary, storing the count of correctness for each labeled sample. $D_{con}$ is the dictionary, storing the count of consistency for each sample.

1: **for** $epoch$ in range(number of epochs) **do**
2:     siterator = iterator($sloader$)
3:     **for** (uninputs, *unlabel-index*) in $uLoader$ **do**
4:         **if** siterator iteration end **then**
4:             siterator = iterator($sLoader$)
4:             update $D_{corr}, D_{con}$
4:             inputs, targets, *label-index* = next($sLoader$)
5:         **else**
5:             inputs, targets, *label-index* = next($sLoader$)
6:         **end if**
7:         $corr_{label}$ = Normalize($D_{corr}[label\text{-}index]$) // We use Min-max normalization here
8:         $con_{label}$ = Normalize($D_{con}[label\text{-}index]$) // Extracting training consistency for batch labeled samples
9:         $con_{unlabel}$ = Normalize($D_{con}[unlabel\text{-}index]$) // Extracting training consistency for batch unlabeled samples
10:        $con_{batch}$ = Concat($con_{label}, con_{unlabel}$)
11:        labeloutput = $model$(inputs)
12:        unlabeloutput = $model$(uninputs)
13:        $output_{batch}$ = Concat(labeloutput, unlabeloutput)
14:        $\mathcal{L}_{CE} = \mathcal{L}_{CE}$(labeloutput, targets)
15:        $\mathcal{L}_{corr} = \mathcal{L}_{corr}($ softmax(labeloutput), $corr_{label}$)
16:        $\mathcal{L}_{con} = \mathcal{L}_{con}$(softmax($output_{batch}$), $con_{batch}$)
17:        $Loss_{total} = \mathcal{L}_{CE} + \lambda_1 \mathcal{L}_{corr} + \lambda_2 \mathcal{L}_{con}$
18:        Update $model$ weights
19:    **end for**
20: **end for**

---

Table 7: Anomaly detection results on 10% (5000) labeled CIFAR10

| Method | Accuracy↑ | AUROC ↑ | Detection error↓ |
|---|---|---|---|
| | CIFAR10 | | |
| Cross-entropy | 0.797±0.002 | 0.751±0.002 | 0.301±0.000 |
| CRL | 0.807±0.003 | 0.762±0.002 | 0.293±0.001 |
| **ours** | **0.818±0.003** | **0.786±0.001** | **0.276±0.001** |

consistency information. We show that our consistency event has superior performance in estimating confidence in terms of the criterion of ordinal ranking. As shown in Fig. 6, compared with maximum softmax output, training consistency estimate has much better performance in distinguishing correct predictions from wrong ones on unlabeled samples. This indicates training consistency has the ability to extract valid confidence information without the necessity of ground truth labels. Furthermore, we also find our consistency event captures the essential of training consistency in a better manner, resulting in better performances across all shown epochs. Though Correctness also achieves good results, the demand of annotations restricts its application in scenarios with limited labeled data. For the rest of this paper, training consistency is especially referring to our confidence estimate method.

A.9 RESULTS ON CANCER SURVIVAL DATASET

In Cancer Survival dataset Liu et al. (2022), histopathological features are used to predict the 5-year survival of lung cancer patients, which can be taken as a classification task. This dataset has 1512 whole slide images (1203 for training, 151 for validation, 158 for testing), 352 of which died in 5

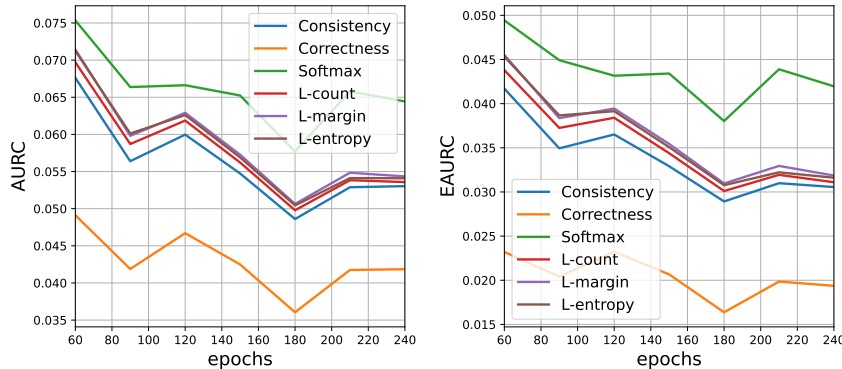

Figure 6: Performance of consistency as a confidence estimator. We evaluate the performance of consistency, correctness, softmax, L-count, L-margin, L-entropy in terms of popular metrics, AURC and E-AURC. We train a model with only 20% of CIFAR10 labeled data, and evaluate on the remaining unlabeled data.

Table 8: Results on Cancer Survival dataset (120 labeled sample)

| Method | Acc↑ | AURC↓ | E-AURC↓ | FPR-95↓ | ECE↓ | NLL↓ | Brier↓ |
|---|---|---|---|---|---|---|---|
| Cancer Survival | | | | | | | |
| CE | 0.616 | 418.50 | 333.03 | 90.53 | 13.76 | 7.35 | 51.86 |
| MMCE (Kumar et al., 2018) | 0.625 | 436.54 | 355.43 | 89.88 | 8.86 | 6.74 | 48.01 |
| CaPE(bin) (Liu et al., 2022) | 0.614 | 428.10 | 341.29 | 92.82 | 9.60 | 7.10 | 50.57 |
| CaPE(kernel) (Liu et al., 2022) | 0.614 | 414.27 | 327.46 | **88.96** | 10.11 | 7.12 | 50.56 |
| Deep ensemble (Lakshminarayanan et al., 2017) | 0.620 | 386.71 | 303.07 | 90.41 | 10.33 | 7.05 | 49.54 |
| ETS (Zhang et al., 2020) | 0.627 | 384.90 | 304.29 | 90.89 | 15.13 | 7.69 | 52.63 |
| Mcdrop | **0.628** | 351.36 | 271.38 | 91.51 | 10.66 | 6.80 | 48.40 |
| CRL | **0.628** | 367.50 | 287.58 | 95.25 | 9.22 | 6.82 | 48.44 |
| Ours | 0.627 | **343.73** | **263.20** | 91.52 | **6.43** | **6.68** | **47.42** |

years. For our experiment, we randomly choose 120 training samples as labeled and the rest training samples are unlabeled.

Results are shown in the Tab. 8. Our method achieves very good performance on this Cancer Survival dataset. Most evaluation metrics of our method are best among compared methods. This is because our method can make use of the confidence information on unlabeled samples effectively.

