# OpenReview forum: "Confidence Estimation Using Unlabeled Data"
_ICLR.cc/2023/Conference — ICLR 2023 poster_

### Official Review · Reviewer_L49h · 2022-10-18

**Confidence:** 4
**Clarity, Quality, Novelty And Reproducibility:** The paper is well written. Clarity an…
**Correctness:** 4
**Technical Novelty And Significance:** 4
**Empirical Novelty And Significance:** 4
**Recommendation:** 8

**Strength And Weaknesses:**

Strengths:
- They take advantage of unlabelled sampled. Having unlabelled samples is a common situation in practical problems and especially very natural situation in active learning problems.
- Numerical results looks good: they reach state-of-art in most of experiments, often with a large margin
- They also provide theoretical considerations supporting their approach

Weaknesses:
- It seems that the approach is only applicable with networks with softmax output (i.e. classification or segmentation), not clear if can be extended to regression etc?
- It is not clear to me that what are actually the metrics used in active learning methods. I guess "entropy" results is computed from softmax output? "MC-dropout" is also based on entropy? Your method is based on softmax or entropy?

Minor weaknesses / comments :
- Baselines are based on quite old studies (the latest is from 2018). There are also several more recent approach. But on the hand, MCdropout is still one of best performing solutions and none are considering semisupervised setup, so this is also just a minor comment.
- Furthermore, numerical results considers only a single set such that training and evolution sets are quite similar (i.e. I would say those are "same distributions"). Some confidence studies considers out-of-distribution detection, meaning that they study if method can detect a sample which is from a significantly different distribution (e.g. Dirty MNIST introduced by Mukhoti & Gal, https://paperswithcode.com/dataset/dirty-mnist). I would like to see also this aspects of their method. But can also be future study.
- Another similar study could be if method can separate aleotoric (uncertainty due to noise etc) and epistemic uncertainty (uncertainty due to lack of training data etc). This is meaningful in active learning if different levels of aleatoric noise are present as it is pointless to pick samples with high aleotoric noise. See Yarin Gal's work. But can also be future study.

Questions:
- Double sum consistency loss is expensive to compute as the authors mention. But can it be avoided? Does the loss sense make any sense without taking only difference of consistency and softmax output  (i.e. computing $\sum_s\max\{0,c_s-\kappa_s\}$?
- In training, does minibatch include fixed ratio of labelled and unlabelled samples or is this ratio random? If latter, is it a problem if no labelled samples end up to the mini batch?


**Summary Of The Paper:**

Authors propose a new method for uncertainty estimation in image classification and segmentation considering a semisupervised setting. Their approach takes advantage of also unlabelled data, which is a novel aspect in uncertainty estimation. They propose so called consistency loss which forces softmax output to consistence (defined as consistency of predictions between training epochs). The proposed approach is validated numerically using CIFAR10&100, image segmentation (ISIC2017) and in a active learning experiment.

**Summary Of The Review:**

The approach seems novel and relatively simple, and numerical experiments also looks very promising. There are a few points that I would have liked to be see in their study (see "Minor weaknesses / comments / questions" above), but on the hand, I agree that not all can be included into this paper and those could also be part of future study etc. Therefore I would recommend acceptance.

---

> ### Author Response · Authors · 2022-11-19
> **Response to reviewer L49h (1/2)**
>
> Thank you for your detailed evaluation and constructive suggestions. We also appreciate that you find our work novel and promising. We hope to address your concerns with additional experiments and the following discussions.
>
> **Q1**: Only softmax for classification, can expand to regression?
>
> **A1**: Good question. For regression, there could be a straightforward generalization of our consistency measure: the difference between model outputs at two consecutive epochs. Lower aggregated difference indicates higher consistency and leads to higher confidence on certain samples. This will be an interesting future work for exploration.
>
> **Q2**: It is not clear to me what are actually the metrics used in active learning methods.
>
> **A2**: We are not 100% sure what you meant. In active learning experiments, we use maximum softmax output as the confidence metric and classification accuracy as the evaluation metric. If this doesn’t cover your question, could you please be more specific?
>
> **Q3**: "MC-dropout" also based on entropy?
>
> **A3**: Yes. For ‘MC-dropout’, We calculate 50 stochastic predictions and use the mean of these predictions as the output, and the entropy of the mean softmax output is used as the confidence output of ‘MC-dropout’.
>
> **Q4**: Your method is based on softmax or entropy?
>
> **A4**: Our result is based on maximum softmax output. In the final version, we will be more clear about these questions.
>
> **Q5**: Baselines are quite old studies.
>
> **A5**: We have added newer baselines: CaPE(bin) (2022), CaPE(kernel) (2022), ETS (2020), MMCE (2018). Please refer to Q1 of general response for more details.
>
> **Q6**: Out-of-distribution detection.
>
> **A6**: Thanks for the suggestion. We run out-of-distribution experiments on CIFAR10 (CIFAR100) and Dirty MNIST.
>
> We evaluate the anomaly detection ability of our method on CIFAR10 and use the samples of CIFAR100 as out-of-distribution samples. In this part, we use the 10% (5000) labeled samples setting. The results are shown in the table below, and our method achieves best performance. This indicates that the deep classifier trained by our method can detect out-of-distribution samples efficiently. For the training details please refer to the main paper Sec 4.1. We compare our method with Cross-entropy loss and the correctness ranking loss. Here we use detection error (the minimum error among all thresholds) and the area under the receiver operating characteristic curve (AUROC) to evaluate the out-of-distribution performances. All experiments are iterated 3 times. The mean and standard deviation are reported.
>
> *Table: Out-of-distribution results on CIFAR10*
> | Method        | Classification Accuracy $\uparrow$ | AUROC $\uparrow$       | Detection error $\downarrow$ |
> |---------------|---------------------------|-----------------|-------------------|
> | Cross-entropy | 0.797$\pm$0.002               | 0.751$\pm$0.002     | 0.301$\pm$0.000       |
> | CRL           | 0.807$\pm$0.003               | 0.762$\pm$0.002     | 0.293$\pm$0.001       |
> | Ours          | **0.818$\pm$0.003**           | **0.786$\pm$0.001** | **0.276$\pm$0.001**   |
>
> We also use Dirty MNIST introduced by [5] to verify our method in out-of-distribution detection. We train our model on 10% labeled and 90% unlabeled clean MNIST samples. We evaluate on 10000 clean test samples and 60000 ambiguous samples (out-of-distribution). As shown in the table below, our method achieves good performance.
>
> | Method        | Classification Accuracy $\uparrow$ | AUROC $\uparrow$       | Detection error $\downarrow$ |
> |---------------|---------------------------|-----------------|-------------------|
> | Cross-entropy | **0.987$\pm$0.002**           | 0.913$\pm$0.005     | 0.091$\pm$0.003       |
> | CRL           | 0.985$\pm$0.002               | 0.918$\pm$0.006     | 0.072$\pm$0.005       |
> | Ours          | 0.982$\pm$0.002               | **0.927$\pm$0.000** | **0.067$\pm$0.001**   |
>
> [5] Mukhoti, Jishnu, et al. "Deep Deterministic Uncertainty: A Simple Baseline." arXiv e-prints (2021): arXiv-2102.

---

> > ### Author Response · Authors · 2022-11-19
> > **Response to reviewer L49h (2/2)**
> >
> > **Q7**: If our method can  separate aleatoric uncertain samples and epistemic uncertain samples.
> >
> > **A7**: Our assumption is that if our method can separate aleatoric and epistemic uncertain samples, then we expect the model trained by our model to assign lower confidence to epistemic uncertain samples and higher confidence to aleatoric uncertain samples. But it is tricky to know the uncertainty type for specific samples. Although we can generate aleatoric uncertain samples by adding noise to normal images and epistemic uncertain samples by introducing samples from other dataset, it is not easy to generate aleatoric and epistemic uncertain samples with comparable confidence level. Therefore, it would be difficult to observe our model performance on distinguishing aleatoric from epistemic samples. But still, this question deserves further study. Please kindly let us know if you have any specific advice about experiment verification. This is a very interesting topic worth exploring.
> >
> > **Q8**: Double sum consistency loss is expensive to compute as the authors mention. Does the loss make any sense without taking only the difference of consistency and softmax output (i.e. computing $ \sum_s |c_s - \kappa_s| $)?
> >
> > **A8**: Great question. Unfortunately, training consistency is not of the same scale as the probability-based uncertainty. Thus pointwise loss like L1 would not work. Instead, we use consistency ranking loss so that only the ranking of consistency is learnt. We conduct an ablation study using L1 loss to substitute ranking loss. As shown in the table below, the L1 loss (‘pointwise L1’ row) has much worse performance than ranking loss in confidence estimation.
> >
> > *Table: The L1 loss ablation study results on CIFAR10 (5000)*
> > | Method         | Acc $\uparrow$            | AURC $\downarrow$           | E-AURC $\downarrow$         | FPR $\downarrow$           | ECE $\downarrow$          | NLL $\downarrow$           | Brier $\downarrow$          |
> > |----------------|-----------------|----------------|----------------|----------------|---------------|---------------|----------------|
> > | Pointwise (L1) | 0.803$\pm$0.004     | 76.733$\pm$5.79    | 55.88$\pm$4.87     | 66.97$\pm$0.75     | **3.25$\pm$0.71** | 7.84$\pm$0.64     | 28.51$\pm$0.71     |
> > | ours           | **0.818$\pm$0.002** | **44.43$\pm$1.03** | **26.88$\pm$0.66** | **64.64$\pm$0.93** | 5.26$\pm$0.43     | **6.42$\pm$0.08** | **26.88$\pm$0.43** |
> >
> > About the efficiency concern, we found that only with limited pairing in mini-batch, our method can achieve very good performance. Details about the pairing strategy we use are in Appendix Section A.2.
> >
> > **Q9**: In training, does minibatch include a fixed ratio of labeled and unlabelled samples or is this ratio random? If the latter, is it a problem if no labeled samples end up in the mini batch?
> >
> > **A9**: We use a fixed ratio of labeled and unlabelled samples within each mini-batch. More details about labeled and unlabelled samples ratios in mini-batch can be seen in Sec 4.

---

> > > ### Comment · Reviewer_L49h · 2022-11-22
> > > **Satisfied with the response**
> > >
> > > I am generally satisfied the response and new results are interesting. I keep my rating unchanged as it was already pointing to clear acceptance.
> > >
> > > Response to question from the authors:
> > > Q2/A2: I meant that what was the metric or strategy to pick samples to be labelled. I guess it is "maximum softmax output" based on A4. For example, Yarin Gal et al had paper ("Deep Bayesian Active Learning with Image Data") in which they compared different metrics.
> > >
> > > A7: Yarin Gal's team has studies this topic (I think also in your [5] if I remember correctly), you can perhaps follow their examples?

---

> > > > ### Author Response · Authors · 2022-11-26
> > > > **Further clarification**
> > > >
> > > > Dear Reviewer L49h,
> > > >
> > > > We are glad that you are satisfied with our response and we would like to provide further clarifications to strengthen our claims.
> > > >
> > > > **Q2**: It is not clear to me what are actually the metrics used in active learning methods.
> > > >
> > > > **A2**: Yes we use "maximum softmax output" for our method in the active learning section.
> > > >
> > > > **Q7**: If our method can separate aleatoric uncertain samples and epistemic uncertain samples.
> > > >
> > > > **A7**: Based on Yarin Gal team’s work, deterministic CNN with acquisition functions like softmax captures aleatoric uncertainty rather than epistemic uncertainty. We compare our method with the deterministic model using the same acquisition function (softmax). As shown in table below, our method performs better in all active learning stages than deterministic model. This indicates that the high uncertainty samples acquired by our method can help active learning in a better manner and demonstrates that our method can separate epistemic from aleatoric uncertainty. For experiment setting, we follow the Sec. 4.3 of the main paper.
> > > >
> > > > (*Table: active learning results by comparing with deterministic models (accuracy)*)
> > > > |              | Stage 0         | Stage 1         | Stage 2         | Stage 3         | Stage 4         | Stage 5         | Stage 6         | Stage 7         | Stage 8         | Stage 9         |
> > > > |------------------------|-----------------|-----------------|-----------------|-----------------|-----------------|-----------------|-----------------|-----------------|-----------------|-----------------|
> > > > | Softmax      | 0.509$\pm$0.033     | 0.657$\pm$0.011     | 0.740$\pm$0.022     | 0.791$\pm$0.005     | 0.817$\pm$0.004     | 0.846$\pm$0.003     | 0.863$\pm$0.003     | 0.874$\pm$0.003     | 0.884$\pm$0.002     | 0.889$\pm$0.002     |
> > > > | Ours-softmax | **0.553$\pm$0.011** | **0.704$\pm$0.012** | **0.778$\pm$0.012** | **0.829$\pm$0.008** | **0.850$\pm$0.006** | **0.869$\pm$0.004** | **0.881$\pm$0.004** | **0.889$\pm$0.004** | **0.897$\pm$0.005** | **0.902$\pm$0.006** |

---

### Official Review · Reviewer_qWap · 2022-10-24

**Confidence:** 4
**Correctness:** 3
**Technical Novelty And Significance:** 2
**Empirical Novelty And Significance:** 3
**Recommendation:** 6

**Clarity, Quality, Novelty And Reproducibility:**

The paper is almost clear and seems reproducible. However, the novelty and the scientific quality in terms of contributions are limited.


**Strength And Weaknesses:**

Strength:

-The problem of overconfident predictions is important and requires major attention from the community.
-The paper is well-written and easy to understand.
-Extensive experiments are conducted to evaluate the performance of the model.



Weaknesses:
MAJOR:
-In Figure 1, the paper attempts to show that the training consistency correlates with the distance to the decision boundary and then claims that the distance to the decision boundary can be interpreted as confidence. This claim does not seem to be correct. Since the output space is not calibrated, the distance to the decision boundary in the output is also overconfident and cannot be used to represent confidence. We can see this more clearly by noting that the linear classifier that performs the classification on top of the representations is also estimating a relative distance to the decision boundary. Therefore, the overconfidence problem would not exist in the first place if the distances in the representation space were correlated with confidence.
-Another point is that the distance in the output may correlate with confidence only on the samples that are close to the decision boundary as depicted in Figure 1. This is obvious because we also observe in conventional networks that low predicted probability shows low confidence. The challenge is actually on samples that are distant from the decision boundary. Therefore, the assumptions made based on the observation in Figure 1 are not accurate and only consider the samples close to the boundary while there is no clue or intuition for samples not close to the boundary.
-The novelty is limited. The main idea is the simple adaptation of the correctness measure proposed by (Moon et al., 2020) to the semi-supervised training setup. Other choices for the adaptation could be considered to augment the contribution.
-L_{corr} in Equation 3 is not defined.


**Summary Of The Paper:**

The paper proposes an approach to estimate the confidence of the predictions using unlabeled data in the semi-supervised setup. The main idea is that training consistency, i.e., the frequency of training data getting the same prediction in sequential training epochs, can serve as a surrogate for the model’s prediction.


**Summary Of The Review:**

The novelty and contribution of the paper are limited. In addition, assumptions made based on the exploratory evaluations are not accurate and require further justification.

---

> ### Author Response · Authors · 2022-11-19
> **Response to reviewer qWap**
>
> Thank you for the careful reading of our paper and detailed feedback. We hope we can address your concerns with the following discussions.
>
> **Q1**: Major issue - Figure 1, the distance to the decision boundary may not be calibrated and thus cannot be interpreted as confidence.
>
> **A1**: Thank you for pointing this out. We agree that the distance is not necessarily well calibrated everywhere. We would like to argue that (1) this figure still supports our statement that **consistency is a good surrogate for confidence**; (2) this figure is merely used for visual illustration. Distance was never used in either our algorithm or our theoretical analysis (Thm. 1). Let us elaborate below.
>
> * In Figure 1, despite the unknown calibration quality of the distance, the key is to compare with the “pseudo-ground truth confidence”, i.e., the correctness measure (Fig. 1(c)). The correctness measure (Moon et al. 2020) is well accepted as a high quality approximation of the confidence. We show that the proposed consistency measure (Fig. 1(d)) is distributed very similar to the correctness measure. Meanwhile, the maximum softmax output (Fig. 1(b)) tends to be overconfident except for near the decision boundary. An additional note is that in Fig. 1(c), the correctness measure is reasonably well correlated with distance from the decision boundary. This suggests that the distance is calibrated reasonably, according to the “pseudo-ground truth confidence”.
>
> * Finally, we stress that Fig. 1 is merely a visual illustration to provide intuition. Distance was neither used in our method/loss nor relevant to our theoretical result (Thm. 1). So even if in the worst case Fig. 1 does not support our statement (which is unlikely as explained above), we still have strong empirical evidence to support our statement.
>
> **Q2**: Novelty is limited: only simple adaptation of the correctness measure (Moon et al. 2020)
>
> **A2**: Our method is novel and nontrivial. Adapting the correctness measure idea from fully supervised setting to semi-supervised setting is not straightforward. The design of our consistency measure, although seems natural, is the outcome of many trial-and-errors and insights gained through careful study of the data through different training stages. We explored many alternative ideas based on prediction frequency, margin, entropy, etc (further explained in question Q3 below).
>
> Finally, we stress that the simplicity of our idea is indeed a strength. It does not have a sophisticated neural network module, depends on very few hyperparameters, is backbone-agnostic, and naturally works on different datasets. This is evident in the strong empirical results, as commented by other reviewers ZHpU and L49h.
>
> **Q3**: Other choices for the adaptation could be considered to augment the contribution.
>
> **A3**: Thanks for the suggestion. We have added some alternative adaptation strategies: prediction maximum label frequency, label frequency entropy and label frequency margin. We compare our strategy with these alternatives and the results are shown in Fig. 6 in the updated manuscript. For a sample, the frequency of model prediction on certain labels during the training process can reflect the model’s confidence level. We can use the maximum label frequency (L-count) as a confidence estimation. Apparently, the higher maximum label frequency introduces less label changes in the training process, leading to higher confidence. Besides, we can also use the label frequency entropy (L-entropy) and margin (L-margin) to summarize consistency information. As shown in Fig. 6 of the main paper, our consistency event has superior performance in estimating confidence in terms of the criterion of ordinal ranking. This indicates that our consistency event captures the essential of training consistency in a better manner, resulting in better performances across all shown epochs. An introduction with more details can be seen in our main paper Appendix Section A.7.
>
> **Q4**: $L_{corr}$ in Equation 3 is not defined.
>
> **A4**: Thanks for pointing this out. We will add this to the main paper in the final version, this is just the correctness ranking loss (Moon et al. 2020). But only applied on limited labeled training data.

---

> > ### Comment · Reviewer_qWap · 2022-11-24
> > **Response to authors**
> >
> > The authors provided a good attempt to address the comments. However, the concerns regarding the validity of the distance to the decision boundary in the output domain for building the approach and the novelty of the method still call for further evaluations and development. I encourage the authors to consider the comments, enhance the manuscript and resubmit.

---

> > > ### Author Response · Authors · 2022-11-25
> > > **Follow-up**
> > >
> > > Dear Reviewer qWap,
> > >
> > > Thanks very much for your timely reply. We would appreciate if you could kindly let us know what kind of evaluation we could provide to better address your concerns regarding distance.
> > >
> > > Best,
> > >
> > > Authors of paper #3422

---

> > > > ### Comment · Reviewer_qWap · 2022-12-07
> > > > **Response to authors**
> > > >
> > > > - The claim at the end of page 2: “We observe that the data further from the decision boundary have higher consistency, and the ones closer to decision boundary have lower consistency.” is just rephrasing the traditional way of measuring consistency which is not accurate as stated in the Definition part of Section 2.
> > > >
> > > > - It is intuitive that the distance to the decision boundary correlates with consistency since samples closer to the boundary are more prone to be placed in another decision region after training. This interprets that the claim “consistency is a good approximation of the confidence” is not accurate enough since the goodness of this approximation is not studied sufficiently. Based on the experimental evaluations on the use of consistency for the semi-supervised task, I agree that consistency can be a “better” approximation than the softmax predictions for confidence. However, the extent of this goodness is not analyzed and its optimality is highly under question without theoretical evidence.
> > > >
> > > > - Another major shortcoming is in the design of Equation 2. The difference between the consistency of the two samples (c_s-c_i) is compared to the difference in their maximum softmax values (k_s-k_i). However, these two quantities do not scale similarly although their range might be the same. Therefore, a more general case of the loss could be defined as G_c(c_s-c_i)-G_k(k_s-k_i) where G_c and G_k are monotonical functions and compensate for the difference in scaling consistency vs softmax probability. Even a simplified version of the two functions can be designed by considering G_c to be a linear mapping and G_k to be the identity function. Therefore, a more general version of the loss could be a(c_s-c_i)+b-(k_s-k_i) where “a” and “b” are scalers. Then the paper could conduct ablation studies to find the optimal values of a and b which potentially could lead to better performance. This again suggests that the current method is not relatively optimal and requires major modification to assure that the functionality is near optimal.
> > > >
> > > > - However, according to the response provided by the authors, I am willing to increase the rating score to 6. This can be increased more if the authors can improve the loss formulation and conduct ablations to find the optimal compensating functions.

---

> > > > > ### Author Response · Authors · 2022-12-09
> > > > > **Thanks very much for your positive feedback**
> > > > >
> > > > > Dear Reviewer qWap,
> > > > >
> > > > > Thanks for your insightful suggestions. We hope to address your concerns with the following discussions.
> > > > >
> > > > > **Q1**: The claim at the end of page 2: “We observe that the data further from the decision boundary have higher consistency, and the ones closer to decision boundary have lower consistency.” is just rephrasing the traditional way of measuring consistency which is not accurate as stated in the Definition part of Section 2.
> > > > >
> > > > > **A1**: You are somehow correct. Here we introduce the intuition behind using training consistency to estimate confidence, and we will reword this sentence to make it more accurate in the final version:  “During the training process, the decision boundary keeps oscillating and the samples closer to the decision boundary are more sensitive to decision boundary changing. Thus, the data further from the decision boundary have higher consistency, and the ones closer to the decision boundary have lower consistency.”
> > > > >
> > > > > **Q2**: However, the extent of this goodness is not analyzed and its optimality is highly under question without theoretical evidence.
> > > > >
> > > > > **A2**: Thanks for your insightful comments. We will reword our claim to ‘consistency is a reasonable approximation of confidence’. So far we use empirical results to prove the superiority of training consistency in confidence estimation. We agree that further theoretical work is needed for exploring goodness and optimality. We will leave this as future work.
> > > > >
> > > > > **Q3**: A more general version of the loss could be $a(c_s-c_i)+b-(k_s-k_i)$ where $a$ and $b$ are scalers. Then the paper could conduct ablation studies to find the optimal values of a and b which potentially could lead to better performance.
> > > > >
> > > > > **A3**: Thanks for your suggestions. We agree that a more generalized formulation of the loss can potentially lead to better performance and we will be happy to explore it in the future.
> > > > >
> > > > > We hope the discussions above have addressed your concerns, and please kindly let us know if you have further concerns.
> > > > >
> > > > > Best,
> > > > > Authors of paper #3422

---

### Official Review · Reviewer_ZHpU · 2022-10-26

**Confidence:** 4
**Correctness:** 4
**Technical Novelty And Significance:** 3
**Empirical Novelty And Significance:** 3
**Recommendation:** 6

**Clarity, Quality, Novelty And Reproducibility:**

The method is novel and clearly explained. I was not able to find the code in the supplementary material or the main paper. Therefore, authors are encouraged to share the code for reproducibility purposes.

**Strength And Weaknesses:**

Strength(s):
1. Using consistency as a surrogate for the correctness and the model's confidence is an interesting and novel idea. The experiment demonstrating a high correlation between consistency and confidence was quite convincing.
2.  The authors nicely transferred their learning about consistency being a decent surrogate to confidence into a loss function that enables softmax outputs to be as close to consistency as possible.
3. Their method can leverage unlabeled data and thereby works well in limited label settings. This is again novel as prior works have mainly focussed on the supervised training setting.
4. Their experiment with active learning was again a clever way to demonstrate their method's superiority.

Weakness(es)/Suggestion(s):
1. It seems that results on CIFAR10 and CIFAR 100 should be supported by a real-world dataset(s) where we actually see some uncertainty in the outcomes like predicting the weather or predicting a future event from the current data.  Recent work in deep probability estimation actually addresses this issue and also proposes some datasets [1]. I feel the proposed method is an interesting one and therefore, should be tried on some real-world uncertainty prediction datasets. It would only strengthen the paper.
2. Additionally, some existing methods like [1], [2], [3], and [4] for the model calibration should be added to the comparisons. This again would strengthen the paper. If you feel that the above methods would obviously not outperform the proposed method, the reasons for the same should be included in the relevant work section.
3. For figure 3, some details like which dataset was used, how many training samples were labeled, etc. are missing.

[1] Deep probability estimation: https://arxiv.org/pdf/2111.10734.pdf
[2] Deep ensembles: https://arxiv.org/pdf/1612.01474.pdf
[3] Mix and Match: https://arxiv.org/pdf/2003.07329.pdf
[4] MMCE: https://proceedings.mlr.press/v80/kumar18a/kumar18a.pdf


**Summary Of The Paper:**

The paper introduces a consistency ranking loss that enables the model's softmax output to be a close approximation of consistency. Consistency in turn correlates well with confidence estimation. In other words, if the model's prediction for a sample is consistent as the training progresses, it is likely to have high confidence in its prediction for that sample. Their method can leverage unlabeled data and thereby works well in limited label settings. They show that their method of estimating confidence outperforms other methods for image classification and image segmentation tasks.

**Summary Of The Review:**

The method proposed in the paper tackle many important issues like confidence estimation in semi-supervised settings which was not adequately addressed in the previous work. Additionally, the proposed method is sound, novel, and elegant. Therefore, my recommendation would be marginally above the acceptance threshold (6).

---

> ### Author Response · Authors · 2022-11-19
> **Response to reviewer ZHpU**
>
> We appreciate the reviewer for pointing out specific datasets, baselines, as well as presentation issues that could be improved. We have added additional experiments and improved our manuscript according to the constructive suggestions.
>
> **Q1**: Some empirical datasets in a certain paper are recommended for strengthening the paper.
>
> **A1**: We have added one of the recommended datasets. Please refer to Q1 of general response.
>
> **Q2**: 4 more baseline methods for comparison (or discussion).
>
> **A2**: We have added these baselines. Please refer to Q1 of general response.
>
> **Q3**: Some details for Figure.3 are missing.
>
> **A3**: We are using labeled sample size 250 on the ISIC2017 dataset. The experiment setting is similar to Sec 4.2 (medical image segmentation). The only difference is the weight on consistency ranking loss. Thanks for pointing this out, we will add them to the main paper in the final version.

---

> > ### Comment · Reviewer_ZHpU · 2022-12-08
> > **Thank you for your response**
> >
> > Thank you authors for your response. Since the new results submitted to the reviewers strengthen the paper, I would recommend shifting them to the main paper from the appendix. These new results can include comparisons to the recent baselines, performance on the real-world dataset, ablation studies of the method, pseudo-code, and any additional results on OOD and active learning. If there is a shortage of space then some of the theoretical analysis in the main paper can be moved to the appendix.
> >
> > I maintain my acceptance score of 6.

---

> > > ### Author Response · Authors · 2022-12-09
> > > **Thanks for your support**
> > >
> > > Dear Reviewer ZHpU,
> > >
> > > Thanks for your suggestions. Depending on additional revisions we need to make in the final version, we will try to add more experiment results if space permits.
> > >
> > > Best,
> > >
> > > Authors of paper #3422

---

### Official Review · Reviewer_pST4 · 2022-10-27

**Confidence:** 3
**Correctness:** 4
**Technical Novelty And Significance:** 2
**Empirical Novelty And Significance:** 2
**Recommendation:** 6

**Clarity, Quality, Novelty And Reproducibility:**

Firstly, in related work pg2, in the second sentence from last, there is a missing citation "?".
Secondly, the paper would benefit from removing some redundant formulations, and instead adding more important ones from the appendix. Especially on pages 2,3.
Thirdly, the idea of expanding confidence in semi-supervised learning settings might be kind of novel and original but some ablation studies to understand limitation and stress test it would have been useful.
Reproducibility is at the bare minimum, as it is only the maths available (and might not be everything needed there), and the fact that they have used Resnet-18 for the active learning and other Resnet variants for other parts of the paper. I would have expected to see a pseudo-code/algorithm at the very least.

**Strength And Weaknesses:**

Strengths:

a)	A simple but effective method, that works competitively with respect to other methods.
b)	The authors also show some results in medical image segmentation
c)	In theory, it should generalise well in other datasets

Weaknesses:

a)	I think it is a weak formulation and some quantification of the bias of the estimator would be useful – to be honest, though appendix covers me but it is also not an excuse to use the appendix to write a longer paper
b)	I appreciate the use of medical images but results on more datasets would demonstrate the generalisation of the method, as CIFAR and ISIC2017 might be not the best ones for this task
c)	I think most of the formulas and maths in the first half of the paper, mainly pages 2 and 3 are redundant in my opinion. You want to explain something very simple and the maths just makes it more complicated than it is.



**Summary Of The Paper:**

In a nutshell, authors extend confidence estimation in DNNs to semi-supervised learning. To my understanding, the way they have approached the problem is that the more consistent the prediction for a given data sample x_i across different stages of training (including weights) is, the more confident the model will be; and that is independent of y_i, i.e. consistency is the proxy. To some extent it reminds me of this work (https://link.springer.com/article/10.1007/s00521-019-04332-4) where the authors annotated unlabelled data based on uncertainty estimation. The results presented (and visualised, e.g. figure 1) are promising, but half of the paper (especially the first half) tries to mathematically formulate the problem unnecessarily. Even the theorem might be redundant, with the most relevant being eq. 2.

**Summary Of The Review:**

Good paper, with some useful intuitions around expanding confidence in semi-supervised settings, but the paper lack proper organisation, useful ablations/experiments, and a more compelling provision of the importance of this work. It would require some moderate improvements to make it publishable, but it is nevertheless an important piece of work.

---

> ### Author Response · Authors · 2022-11-19
> **Response to reviewer pST4 (1/2)**
>
> Thanks very much for your detailed comments which are useful for us to improve our manuscript. We appreciate your careful reading and insightful advice in readability. We hope to address your concerns with additional experiments and pseudo-code.
>
> **Q1**: Some quantification of the bias of the estimator would be useful.
>
> **A1**: We are not sure what you are referring to. In the question below (Q7), we provide some ablation study results. We hope they help.
>
>
> **Q2**: Though appendix covers me but it is also not an excuse to use the appendix to write a longer paper.
>
> **A2**: Could you please be specific which part in the appendix you are referring to? We will be happy to move them to the main paper.
>
> **Q3**: More datasets results are necessary.
>
> **A3**: Please refer to the General response Q1 regarding a new dataset.
>
> **Q4**: Formulas are redundant in page 2~3. Provide implementation details from the appendix.
>
> **A4**: On page 2-3, we formalized the definition of consistency (Eq. 1). We also introduced other notations necessary for the definition, such as classifiers, labels, etc. We will try to simplify these notations to improve readability.
>
> **Q5**: Reproducibility: pseudo-code and code.
>
> **A5**: Thanks for your suggestion. We have added the pseudo-code in Appendix Section A.6. And we will release our code upon acceptance.
>
> **Q6**: Missing citation.
>
> **A6**: We have fixed this citation issue.

---

> > ### Author Response · Authors · 2022-11-19
> > **Response to reviewer pST4 (2/2)**
> >
> > **Q7**: Ablation studies to understand limitations and stress tests would have been useful.
> >
> > **A7**: In the original paper, we conducted an ablation study on the weight of the consistency ranking loss (See Sec 4.2 and Figure.3). Our method is not sensitive to this hyperparameter. In addition, we provide two new ablation studies: with/without correctness supervision, normalization strategy. In all studies, we run experiments 3 times and report the mean and standard deviation of the performances. More details of these studies and the results have been added to Appendix Section A.9.
> >
> > * *(with/without correctness supervision.)* We used correctness ranking loss in our method to make full use of limited training labels. Here we conduct an ablation study to see the importance of the correctness ranking loss. According to the tables below, removing correctness ranking loss hurts the performance on both classification and segmentation tasks.
> >
> > *Table: The correctness supervision ablation study results on CIFAR10 (5000 labeled)*
> > | Method            | Acc $\uparrow$             | AURC $\downarrow$          | E-AURC $\downarrow$         | FPR $\downarrow$            | ECE $\downarrow$           | NLL $\downarrow$           | Brier $\downarrow$          |
> > |-------------------|-----------------|----------------|----------------|----------------|---------------|---------------|----------------|
> > | w/o corr          | 0.816$\pm$0.004     | 47.12$\pm$1.87     | 29.16$\pm$1.41     | 66.37$\pm$0.86     | 5.39$\pm$0.36     | 6.74$\pm$0.16     | 27.38+0.32     |
> > | w/ corr (ours)  | **0.818$\pm$0.002** | **44.43$\pm$1.03** | **26.88$\pm$0.66** | **64.64$\pm$0.93** | **5.26$\pm$0.43** | **6.42$\pm$0.08** | **26.88$\pm$0.43** |
> >
> > *Table: The correctness supervision ablation study results on ISIC2017 (250 labeled)*
> > | Method            | mIOU $\uparrow$             | AURC $\downarrow$          | E-AURC $\downarrow$         | FPR $\downarrow$            | ECE $\downarrow$           | NLL $\downarrow$           | Brier $\downarrow$          |
> > |-------------------|-----------------|----------------|----------------|----------------|---------------|---------------|----------------|
> > | w/o corr          | 0.809$\pm$0.004     | 13.79$\pm$0.58     | 10.98$\pm$0.66     | 55.15$\pm$2.02     | 3.36$\pm$0.36     | 2.66$\pm$0.07     | 11.32$\pm$0.16     |
> > | w/ corr (ours)  | **0.817$\pm$0.007** | **12.79$\pm$0.79** | **10.21$\pm$0.62** | **54.51$\pm$2.79** | **2.56$\pm$0.26** | **2.23$\pm$0.17** | **10.71$\pm$0.31** |
> >
> > * *(normalization strategy.)* In the original paper we do min-max normalization for training consistency on labeled samples and unlabeled samples in mini-batch separately. Here we try to concatenate training consistency on labeled and unlabeled samples first than normalize mini-batch consistency together as an alternative solution. The results are shown in the table below. We can see all evaluation metrics deteriorate significantly with the alternative normalization strategy.
> >
> > *Table: The normalization strategy ablation study results on CIFAR10 (5000 labeled)*
> > | Method            | Acc $\uparrow$             | AURC $\downarrow$          | E-AURC $\downarrow$         | FPR $\downarrow$            | ECE $\downarrow$           | NLL $\downarrow$           | Brier $\downarrow$          |
> > |-------------------------------|-----------------|----------------|----------------|----------------|---------------|---------------|----------------|
> > | Unified normalization         | 0.812$\pm$0.001     | 48.03$\pm$2.31     | 29.07$\pm$2.26     | 64.83$\pm$0.79     | 9.34$\pm$0.36     | 7.33$\pm$0.06     | 29.87$\pm$0.02     |
> > | Normalizing separately (ours) | **0.818$\pm$0.002** | **44.43$\pm$1.03** | **26.88$\pm$0.66** | **64.64$\pm$0.93** | **5.26$\pm$0.43** | **6.42$\pm$0.08** | **26.88$\pm$0.43** |

---

> > > ### Comment · Reviewer_pST4 · 2022-11-21
> > > **Very happy with the response**
> > >
> > > Many thanks for additional information provided - very enlightening and helpful.
> > > I would add the A.9 ablation part to the main paper. I am happy to increase my score.

---

> > > > ### Author Response · Authors · 2022-11-21
> > > > **Thanks very much for your positive feedback**
> > > >
> > > > Dear Reviewer pST4,
> > > >
> > > > Thanks very much for your positive feedback. We are happy to add the A.9 ablation part to the main paper in the final version. Again, thanks for your valuable comments to improve our manuscript.
> > > >
> > > > Best,
> > > >
> > > > Authors of paper #3422

---

### Author Response · Authors · 2022-11-19
**General response**

We appreciate all reviewers for their time and insightful comments. We revised the manuscript based on the constructive feedback and suggestions from the reviewers. We have uploaded the revised version to reflect the modifications (highlighted in blue). We add additional experiments regarding concerns of reviewers to improve our manuscript.

**Q1**: More datasets and baselines. (Reviewer pST4, ZhpU, L49h)

**A1**: Reviewer ZhpU asked for results on real-world uncertainty prediction datasets (e.g., Cancer Survival dataset, weather forecasting and Collision Prediction) and suggested some new baselines: MMCE [4], CaPE(bin) [1], CaPE(kernel) [1], Deep Ensemble [2], ETS [3]. Here we compare our methods with these suggested baselines on the suggested Cancer Survival dataset [1]. We will cite and discuss these baselines in the final version.

This also addresses Reviewer pST4’s request for more datasets and Reviewer L49h’s request for more recent baselines.

In Cancer Survival dataset [1], histopathological features are used to predict the 5-year survival of lung cancer patients, which can be taken as a classification task. This dataset has 1512 whole slide images (1203 for training, 151 for validation, 158 for testing). For our experiment, we randomly choose 120 training samples as labeled and the rest training samples are unlabeled.

Results are shown in the table below. Our method achieves very good performance on this Cancer Survival dataset. Most evaluation metrics of our method are best among compared methods. This is because our method can make use of the confidence information on unlabeled samples effectively. We have added the results to Appendix Section A.8.

|     Method    |    Acc $\uparrow$    |    AURC $\downarrow$   |   E-AURC $\downarrow$   |    FPR $\downarrow$  |    ECE $\downarrow$ |    NLL $\downarrow$ |   Brier $\downarrow$ |
|-------------|:---------:|:----------:|:----------:|:---------:|:--------:|:--------:|:---------:|
| CE            | 0.616     | 418.50     | 333.03     | 90.53     | 13.76    | 7.35     | 51.86     |
| MMCE          | 0.625     | 436.54     | 355.43     | 89.88     | 8.86     | 6.74     | 48.01     |
| CaPE(bin)     | 0.614     | 428.10     | 341.29     | 92.82     | 9.60     | 7.10     | 50.57     |
| CaPE(kernel)  | 0.614     | 414.27     | 327.46     | **88.96** | 10.11    | 7.12     | 50.56     |
| Deep Ensemble | 0.620     | 386.71     | 303.07     | 90.41     | 10.33    | 7.05     | 49.54     |
| ETS           | 0.627     | 384.90     | 304.29     | 90.89     | 15.13    | 7.69     | 52.63     |
| Mc dropout    | **0.628** | 351.36     | 271.38     | 91.51     | 10.66    | 6.80     | 48.40     |
| CRL           | **0.628** | 367.50     | 287.58     | 95.25     | 9.22     | 6.82     | 48.44     |
| Ours          | 0.627     | **343.73** | **263.20** | 91.52     | **6.43** | **6.68** | **47.42** |

*(More details about the baselines.)*
* Deep Probability Estimation [1] proposes a calibration loss (CaPE) to promote the model probability estimation in the training process, but this method relies on early-stop models, whose probability estimation can be impaired by lack of training samples.
* Deep ensemble [2] is expensive in both inference time and storage.
* ETS [3] ensembles multiple temperature scaling transformations to improve the performance of temperature scaling transformations in calibration.
* MMCE [4] is a trainable RKHS kernel based calibration measure, which can be trained efficiently with NLL (negative likelihood loss). Same as previous methods, MMCE cannot utilize unlabeled samples efficiently and thus underperforms. We will cite and discuss them in the final version.

*(Experiment setting.)* We train our model for 200 labeled training epochs with batch size 32 (16 labeled). The remaining setting is similar to the setting in our paper (Sec 4.1). The baseline methods: CE (cross-entropy), MMCE [4], CaPE(bin) [1], CaPE(kernel) [1], Deep Ensemble [2], ETS [3] are implemented with the public repo from here: https://github.com/jackzhu727/deep-probability-estimation.

[1] Sheng Liu, Aakash Kaku, Weicheng Zhu, Matan Leibovich, Sreyas Mohan, Boyang Yu, Laure Zanna, Narges Razavian, and Carlos Fernandez-Granda. Deep probability estimation. In ICML, 2022.

[2] Balaji Lakshminarayanan, Alexander Pritzel, and Charles Blundell. Simple and scalable predictive uncertainty estimation using deep ensembles. NeurIPS, 2017.

[3] Jize Zhang, Bhavya Kailkhura, and T Yong-Jin Han. Mix-n-match: Ensemble and compositional methods for uncertainty calibration in deep learning. In ICML, 2020.

[4] Aviral Kumar, Sunita Sarawagi, and Ujjwal Jain. Trainable calibration measures for neural networks from kernel mean embeddings. In ICML, 2018.

Below we address specific concerns one-by-one.

---

> ### Comment · Reviewer_pST4 · 2022-11-21
> **Great additions**
>
> Very good addition to the paper and extensive justification.

---

### Decision · Program_Chairs · 2023-01-20

**Decision:**

Accept: poster

**Justification For Why Not Higher Score:**

The paper has sufficient novelty for poster, but it is not an outstanding one for oral.

**Justification For Why Not Lower Score:**

The idea is interesting and effective. The experimental results, especially after rebuttal with additional results, are sufficient for accepting this paper.

**Metareview: Summary, Strengths And Weaknesses:**

Summary: The paper's main contribution is to use training consistency, i.e., the frequency of a training datum getting the same prediction in sequential training epochs, as a surrogate function of the model’s confidence. While the idea is simple and straight forward, the design of the loss function is non-trivial. This is also the main strength of the paper.
The main weakness of the paper is actually the presentation. Some parts of the paper such as the theoretical analysis are redundant. Instead, the authors are suggested to include those results in the supplementary into the main paper.Ther



**Note From Pc:**

if the above contains the word "oral" or "spotlight" please see: "oral" presentation means -> notable-top-5% and "spotlight" means -> notable-top-25%. As stated in our emails, we are disassociating presentation type from AC recommendations

**Summary Of Ac-Reviewer Meeting:**

There is a consensus that the paper has sufficient novelty and quality for ICLR, but it is also agreed that the paper is not an outstanding one for oral.
The presentation of the paper needs to be improved. Based on the comments from reviewers and agreed by AC, the authors need to include the additional results in the main context while putting the theoretical analysis and some other parts into appendix.